# The Latest Trends in Electric Vehicles Batteries

**DOI:** 10.3390/molecules26113188

**Published:** 2021-05-26

**Authors:** Rui Martim Salgado, Federico Danzi, Joana Espain Oliveira, Anter El-Azab, Pedro Ponces Camanho, Maria Helena Braga

**Affiliations:** 1DEMec, Faculdade de Engenharia da Universidade do Porto, Rua Dr. Roberto Frias, s/n, 4200-465 Porto, Portugal; pcamanho@fe.up.pt; 2LAETA, Engineering Faculty, Engineering Physics Department, University of Porto, R. Dr. Roberto Frias s/n, 4200-465 Porto, Portugal; fdanzi@inegi.up.pt (F.D.); jespain@fe.up.pt (J.E.O.); 3INEGI, Instituto de Ciência e Inovação em Engenharia Mecânica e Engenharia Industrial, Rua Dr. Roberto Frias, 400, 4200-465 Porto, Portugal; 4School of Materials Engineering, Purdue University, 701 West Stadium Avenue, West Lafayette, IN 47907, USA; aelazab@purdue.edu

**Keywords:** batteries for mobility, li-ion batteries, graphite/silicon anodes, traditional cathodes

## Abstract

Global energy demand is rapidly increasing due to population and economic growth, especially in large emerging countries, which will account for 90% of energy demand growth to 2035. Electric vehicles (EVs) play a paramount role in the electrification revolution towards the reduction of the carbon footprint. Here, we review all the major trends in Li-ion batteries technologies used in EVs. We conclude that only five types of cathodes are used and that most of the EV companies use Nickel Manganese Cobalt oxide (NMC). Most of the Li-ion batteries anodes are graphite-based. Positive and negative electrodes are reviewed in detail as well as future trends such as the effort to reduce the Cobalt content. The electrolyte is a liquid/gel flammable solvent usually containing a LiFeP${}_{6}$ salt. The electrolyte makes the battery and battery pack unsafe, which drives the research and development to replace the flammable liquid by a solid electrolyte.

## 1. Introduction

Lithium-ion batteries (LIBs) using Lithium Cobalt oxide, specifically, Lithium Nickel-Manganese-Cobalt (NMC) oxide and Lithium Nickel-Cobalt-Aluminium (NCA) oxide, still dominate the electrical vehicle (EV) battery industry with an increasing market share of nearly 96% in 2019, see Figure 1. The same could be stated about recent LIB applications in Grid Storage Technologies (GSTs), although over the last decade Lithium Iron Phosphate LFP-based chemistries had a significant market share in this type of application. However, the industry is committed to developing Cobalt-free cathodes. During its battery day, Tesla revealed plans to substitute Cobalt with Nickel [1] in their batteries’ cathodes. Without specifying the time necessary for shifting towards 0% Cobalt, it can be assumed that predictions identifying NMC as a dominant cathode technology in 2025 are still applicable [2]. Nonetheless, political, social, health and economic concerns are applying increasing pressure to change this situation, since currently Cobalt can only be mined (in sufficient quantities) in the Democratic Republic of the Congo (DRC), under forced labour and hazardous conditions, high prices, and economically unstable trades [3,4]. It is noteworthy that the cathode typically limits cell, and thus LIB performance, as it possesses a lower capacity than the graphitic anode and is the most expensive material of a LIB, leading to its establishment as a key LIB component and the subject of intensive research. Therefore, an enhancement of the cathode leads to a significant impact at the cell level and on overall battery performance [5,6,7,8].

In recent years, scientific and technological progress in batteries has been largely motivated by the automotive industry and, specifically, by small vehicles for urban transportation. Nevertheless, electric mobility is also associated with recent trends of aerial and maritime applications as well as e-Bikes, electric motorcycles, and others.

According to the European Commission [11], shipping accounts for 2–3% of global greenhouse gas (GHG) emissions, with a forecasted increase of 50–250% by 2050 [12]. However, maritime applications have a market share of less than 1% of the total LIBs market, while Li-based batteries are the most widely used battery type for maritime applications [13]. The difficulty in implementing electric solutions in ships is mainly related to their higher power density, cycle, and calendar life demands, as well as safety requirements. Nevertheless, the number of ships with batteries installed, or on order, more than doubled from 150 ships in 2018 to 314 in 2020, which constitutes a major market leap, indicating that LIBs are reaching an interesting level of maturity for maritime applications [14].

Airbus, Boeing and NASA have targeted aircraft electrification as a crucial research and development topic to address. As in the case of maritime and automobile applications, current aircraft-level projects have chosen Li-ion batteries for energy storage due to their unmatchable energy density when compared with other batteries in the market. A good example is Airbus’ recently launched Zephyr, a High-Altitude Pseudo-Satellite (HAPS). This is "the first unmanned aircraft of its kind to fly in the stratosphere" [15]. The HAPS is powered by Li-ion batteries, with state-of-the-art Silicon nanowires (SiNW) as the anode. However, and most likely, all-electric commercial jets will not be able to substitute their fuel-powered counterparts, at least in the near future [16,17], mainly due to insufficient energy density available in LIBs and any other battery technology currently on the market.

Hybrid solutions might represent a significant contribution to reducing CO${}_{2}$ emissions. Airbus, Rolls Royce, and Siemens have collaborated in the E-Fan X hybrid technology demonstrator program, which involved equipping a BAE Systems Avro RJ100, a regional jetliner, with a hybrid engine (gas turbines requiring fuel, and electric engines requiring LIBs). The project was put to an end before the first flight, due to the Covid-19 crisis [18,19]. Airbus will continue with its research efforts on hybrid aircraft, supported by the company’s facility in Ottobrunn (Germany), E-Aircraft System Test House, the "largest test house dedicated exclusively to alternative propulsion systems and fuels in Europe" [14,20]. The company is aiming to launch its first hybrid aircraft in 2030 [21].

Li-Ion Batteries (LIB) for Mobile Applications 

EVs are becoming serious alternatives to internal combustion engine vehicles. According to the International Energy Agency, electric vehicles accounted for 2.6% of global car sales in 2019. This value increased to 4% in 2020, surpassing the initial estimate of 3% provided by the International Energy Agency (IEA) [22]. According to the agency, the Covid-19 pandemic caused an unbalanced drop in global car sales, more aggravated for non-electric vehicles. In fact, EV sales increased by 40% [23] in 2020. LIBs represent a significant market share of the batteries for EV vehicles. The main reason for this is Li’s high electroactivity since this type of high-power vehicles have substantially high voltage requirements in the range of 400–800 V. LIBs allow for fewer batteries to be associated and battery packs in series to match such high voltages, consequently reducing the internal resistance of the batteries leading to lower heat losses, smaller size components and thus reducing weight and cost.

As for Hybrid Electric Vehicles (HEV), more options are available, since Ni-MH batteries can power this type of vehicles, due to their reduced specific energy and power requirements when compared to EVs. With even less energy, power, and cycle life requirements, internal combustion engine vehicles still have Pb-acid batteries installed owing to this technology’s low cost, safety, and easy replacement [24] given their recycling rate of practically 100%.

## 2. The Positive Electrode (Cathode)

### 2.1. Cathode Chemistry and Microstructure

Generally, commercial LIBs possess one or several types of oxides combined as active material(s). These cathodes can be divided into layered, spinel, and polyanion oxides with layered, spinel, and olivine structure, respectively. All three were first introduced by collaborative research between J. Goodenough and other research groups during the 1980s decade [25]. Even though substantial improvements on all types of cathodes’ energy storage features have been achieved, the structural, thermal, and chemical stability of cathode materials are largely a consequence of their oxide structure [25].

Previously, it was mentioned that LIBs dominate the global market, and the cathode exhibits some of the most determinant characteristics of batteries used in commercially available electric automobiles. Furthermore, all passenger vehicles sold in the European market use batteries with cathodes containing Cobalt. Tesla and Panasonic have developed battery cells with Lithium-Nickel-Cobalt-Aluminium oxide (NCA) as the cathode and all models sold by Tesla on the European market have batteries based on this system. On the other hand, the vast majority of car manufacturers incorporate batteries with Nickel-Manganese-Cobalt oxide as the cathode type, with a clear tendency for the NMC622 ratio (LiNi${}_{0.6}$Mn${}_{0.2}$ Co${}_{0.2}$O${}_{2}$) which reduces the Cobalt content. LG Chem, one of the world leaders in the number of NMC batteries sold [26], shows a clear strategy for reducing cobalt content that consists of developing new cathodes with more favorable ratios (reducing the Cobalt content while maintaining or enhancing the performance). The company is focusing on developing NMC712 (LiNi${}_{0.7}$Mn${}_{0.1}$Co${}_{0.2}$O${}_{2}$), NMC811 (LiNi${}_{0.8}$Mn${}_{0.1}$Co${}_{0.1}$O${}_{2}$), and NCA chemistries for the next generation of electric vehicles [27].

In Table 1, cathode types for LIBs are compared. LFP exhibits the highest temperature above which thermal runaway occurs, and withstands the highest number of cycles before degradation, measured by capacity loss. However, plateau voltage (optimum operational voltage) is significantly lower for this type of cathode, which leads to an increasingly interest in developing NMC cathodes. The latter presents the highest cycle life excluding LFP, besides displaying good experimental capacity. In Figure 2, a qualitative comparison summarizes all these main characteristics of the six types of cathodes.

Other chemistries, such as Lithium-Manganese oxide (LMO), were more significant in the first generation of some EV vehicles, such as the Nissan Leaf and Chevy Bolt [28,29], but it appears that their usage and market significance is decreasing, as these and other manufacturers currently opt for the NMC cathodes.

In Table 2 and Table 3, a description of LIBs installed in several commercial EVs. allows us to gain a clearer insight into the market share between the different cathode types. The numbers presented in Figure 1 reflect the options of most manufacturers, as most adopt the NMC chemistry for the cathode. Furthermore, and excluding NCA and LFP, all remaining vehicles included in Table 2 consist of models released more than 2 years ago, indicating a tendency towards abandoning chemistries different from NMC, NCA, and LFP. In Figure 1, the estimated market share for LFP drops to negligible values after 2020. However, this chemistry type is significantly adopted in the Asian market, and especially in the Chinese market, suggesting that the scope of the prediction is limited to other markets. Alternatively, the use of LFP batteries may rise over the next years, due to their lower price. In fact, it has been suggested that the battery pack of future EVs may be customizable, as users may choose between LFP-a less expensive alternative-and NMC-the best option for performance [31].

### 2.2. Lithium-Iron-Phosphate (LFP) and Lithium-Manganese Oxide (LMO)

Despite their low energy density, LFP batteries exhibit several advantages that enable their application in, namely, mobile motorhomes [28] and vehicles with lower performance requirements [41], such as garbage trucks and electric road sweepers, or commercial EVs. These cells provide high cycle life and reduced risk of thermal runaway [42] have no toxic components, low internal resistance, and high-load handling capability [29]. CATL is the main developer of this type of cathode, supplying several car manufacturers from China-its native country. In 2015, LFP batteries were the most popular for plug-in hybrid electric vehicles (PHEVs) and EVs [43], but over the last five years, NMC surpassed this type of cathode, both in market share and research interest.

Polyanion Oxides (Olivine Structure)

Polyanion oxide chemistry with an olivine structure, as depicted in Figure 3 is the main alternative solution to layered oxides on the market of EVs. LiFePO${}_{4}$, which was discovered by Goodenough and colleagues in 1997, remains the only commercially available cathode based on this chemistry type.

Olivine structures are characterized by stronger covalent bonds between oxygen atoms and remaining elements [25], which improves thermal and chemical stability. It is this fundamental difference on the atomic scale that leads to its higher safety and cycle life. However, the kinetics of electron/Li${}^{+}$ diffusion is intrinsically worse than for layered oxides, and commercial batteries based on this type of oxide rely on the synthesis of small particles coated with carbon [44,45]. M. Armand et al. [46] investigated the state of the art for this type of cathodes and suggested that current research interest in this area is focused on optimizing the carbon coating, as low conductivity seems to be the bottleneck for implementation at the EV battery scale. It is also mentioned that both solution and solid-state coating procedures are the major alternatives for carbon coating, although the former tend to be the most energy-effective and to produce a better surface coating [44,45].

### 2.3. Layered Oxides

The layered structure of Figure 4 was the first successful approach at the design of cathodes for LIBs. Layered oxides, Li${}_{x}$MO${}_{2}$ (M- transition metal) rely on Li intercalation between layers of an oxide, in an octahedral structure that provides enhanced ionic pathways for Li-ion during charge/discharge cycles [47]. These materials generally consist of small particles (at the nanoscale), called primary particles, densely packed into larger spherical particles (microscale), the secondary particles [48].

Wei et al. studied the kinetics of Li-ion diffusion in these materials and concluded that diffusion takes place from one octahedral site to the other by two mechanisms: oxygen dumbbell hopping (ODH) and tetrahedral site hopping (TSH) [49], both depicted in Figure 5, where the letter "M" stands for a transition metal ion. In the former, Li${}^{+}$ travels directly through a molecular structure formed by bonds of oxygen atoms, as depicted in Figure 5a. To establish this mechanism, ionic bonds with two transition metal ions (Mn${}^{+}$ and Ni${}^{+}$), on opposite sides of the oxygen structure, need to be established. For the latter mechanism to occur, delithiation must be in progress since Li-ions travel through a divacancy left by Li diffusion, as can be shown in Figure 5b. The mechanism relies on continuous ionic bonding and debonding between each Li-ion and a nearby transition metal (Ni), as the ion progresses through octahedral sites. Both mechanisms are influenced by (**1**) the size of the site involved (oxygen dumbbell and tetrahedral site, respectively), as well as (**2**) electrostatic interaction between Li-ion and transition metal-cations, below/above, for ODH, and cation directly below, for TSH.

Furthermore, given that TSH requires Li divacancies, ODH dominates at the beginning of charging. However, both mechanisms are only possible if energy activation barriers, associated with ionic bonds between Li and transition metal ions, are transposed. Since ODH will only occur if both ionic bonds are broken (one per each side), TSH is the most favorable mechanism, provided sufficient Li divacancies exists. The authors [49] determined a value of the state of charge (SOC) equal to 1/3 for this to occur.

### 2.4. Lithium Cobalt Oxide, LiCoO${}_{2}$

Dating back to 1980, LiCoO${}_{2}$ was the first chemistry to be successfully implemented in a cathode for LIBs [25,50], and it is still the most widely used chemistry in portable devices. Good ionic and electronic conductivities are key advantages of Co-based cathode; another important feature is the structural stability, which is related with the Co${}^{3+/4+}$ ions redox energy, significantly lower than for Li/Li${}^{+}$ and graphite, leading to good cation ordering and high theoretical capacity (∼274 mAh·g${}^{-1}$[51]), with an operating voltage in the range of 3.8–4.3 V. However, since their electronic band is overlapped with the O${}^{2-}$: 2p band [25] (see Figure 6) instability related with oxygen release limits the practical capacity to ∼150 mAh·g${}^{-1}$, as can be shown in Table 1. This value is only approximately 55% of the theoretical capacity.

### 2.5. Lithium Nickel Oxide, Li${}_{x}$NiO${}_{2}$

The second chemistry proposed for commercial LIBs was Li${}_{x}$NiO${}_{2}$, Lithium-Nickel oxide (LNO), introduced by Dahn et al. [52] in 1990. Besides having a lower cost, Ni-based cathodes lead to higher practical capacities than those based in Co, since Ni${}^{3+/4+\;}$ions do not possess such an overlapping band with O${}^{2}$ as Co-ions. However, Ni${}^{3+}$ tends to be reduced to Ni${}^{2+}$[25], which is thermodynamically unstable and has a similar ionic radius to Li${}^{+}$ [6], thus leading to penetration of Ni in Li layers, causing a shift to spinel-like structure and the formation of inactive phases. Temperature is a driving force for this reaction to occur, rendering the thermal stability of LNO cathodes insufficient for most practical applications. During charging (delithiation), the oxide transforms sequentially from hexagonal (H1), at the highest Li content, to monoclinic, hexagonal (H2), and a final hexagonal phase (H3) occurs at low values of Li [7]. The last transition, H2-H3, leads to severe anisotropic contractions, which subsequently lead to intergranular or intragranular crack initiation. New surfaces are repeatedly formed throughout contraction/expansion cycles, increasing side reactions. Consequently, the irreversibility during cycling and thermal stability render this cathode type inadequate for practical applications, and the production of batteries based on this type of cathode has been replaced by more stable alternatives such as NMC cathodes.

### 2.6. Lithium Manganese Oxide, LiMnO${}_{2}$

Another layered oxide based on a single metal transition with commercial relevance is LiMnO${}_{2}$. The main advantages associated with the use of MnO${}_{2}$ are lower cost and environmental impact of Mn, as well as its relatively high operating voltage (highest plateau voltage, and good thermal stability, only inferior to LFP, according to Table 1). However, it suffers from irreversibility issues, as layered to spinel transitions of the oxide tend to occur during charge-discharge. For this reason, its usage as the active material in EV batteries is limited, as can be shown in Table 2, and restricted to batteries developed before the more recent development of NMC batteries. Nevertheless, Mn ions are chemically stable in the presence of Li${}^{+}$ ions, rendering their use in transition metal oxides valuable.

### 2.7. Lithium Nickel Manganese Cobalt Oxide, NMC, and Lithium Nickel Cobalt Aluminum Oxide, NCA

Presently, the most important layered oxide classes for EV applications are the NMC (Li${}_{x}$ Ni${}_{y}$ Mn${}_{z}$ Co${}_{1-y-z}$ O${}_{2}$) and NCA (Li Ni${}_{x}$ Co${}_{y}$ Al${}_{1-x-y}$) cathodes, as can be shown in Table 2 and Table 3. Different microstructures of the former are depicted in Figure 7 and Figure 8, where the ratio os transition metals is 622 and 811 (*xyz*), respectively. The objective of designing this type of electrode is to obtain the best combination of advantages while mitigating all the disadvantages, associated with each single-metal transition oxide mentioned. Cost tailoring may be achieved by limiting the amount of Co, the most expensive of all the metals. Specifically for NMC, and since Mn${}_{2}$O${}_{4}$ is more stable than NiO${}_{2}$, adding Mn mitigates the irreversibility caused by the chemical instability of Ni-based cathodes [25]. Furthermore, Co provides the necessary structural stability and electrical conductivity for long-term electrochemical performance. Finally, Ni is crucial for achieving high specific capacity, which is made obvious by considering the density of states in Figure 6.

Instead of Mn, NCA cathodes contain Al for increasing chemical stability.

Over the last few years, numerous research studies have been targeting the development of Ni-rich cathodes (usually NMC with Ni ratio of 6 or above) to incorporate them in so-called next-generation LIBs [5,8,53,54,55,56,57,58,59,60,61,62,63] for commercial EVs. Most of the experimental and numerical work cited deal with the adversities of high Ni content, namely, chemical and structural instability, both leading to capacity fading, poor rate performance, and potential decay [59]. To mitigate all these failure mechanisms, a careful, but also scalable and cost-effective design approach is necessary, especially at an early stage of manufacturing design. In Figure 9, the typical process used for obtaining NMC particles is depicted.

#### Problems Associated with Ni-rich LiNi${}_{x}$Mn${}_{y}$Co${}_{1-x-y}$O${}_{2}$ Cathodes

Low ionic conductivity and capacity fading due to structural/chemical degradation are the most critical issues regarding Ni-rich cathodes [56,64]. Zhang et al. identified key degradation mechanisms of Ni-rich NMC cathodes [65] divided into different scales: on the micrometer scale, high strain induced by preparation and first cycles lead to microcracking of secondary particles; on the atomic scale, significant volume changes accompanied by high strain states occurring during delithiation/lithiation. Mechanical stresses thus originated create a driving force for cracks to initiate and propagate [66]. To study the capacity fading of Ni-rich cathodes, Ryu et al. studied several NMC cathodes, with 0.6 ≤ x ≤ 0.95 [67]. A significant increase in capacity degradation has been observed for x > 0.8. The authors attributed this to the anisotropy, both in shrinkage during delithiation, and expansion during discharge, caused by phase transition. For lower contents of Ni, this phase transition is inhibited, thus improving structural stability [56,64].

A different type of chemical instability arises from charge/discharge cycles, which lead to the formation of transition metal ions, Ni${}^{4+}$, Mn${}^{3+/4+}$, and Co${}^{4+}$, producing detrimental substances that degrade structural stability and decrease the number of active materials. Consequently, a transition from layered to spinel-like structure and eventually to rock-salt phase emerges which favors electrode/electrolyte chemical activity, such as the reaction of organic electrolytes with free oxygen atoms released by the reduction of Ni${}^{4+}$[57]. Regarding the former consequence, Chen et al. [63] stated that the high capacity of Ni oxides causes severe volume changes and larger vacancies from Li layers, when the battery is charged, allowing for transition metals (Ni, Co, and Mn for NMC) to fill Li layers. Consequently, an initial layered transitional oxide, with transition metals occupying octahedral sites in their respective layers, move to Li layers, creating a rock-salt phase that obstructs Li${}^{+}$ diffusion. On the other hand, side reactions between electrode and electrolyte increase the thickness of the cathode electrolyte interphase (CEI), which, in turn, decreases ionic conductivity and erodes the surface of the cathodes [66]. In another study focusing on anti-site defect-induced intragranular cracking [61], NMC811 cathodes were investigated with scanning transmission electron microscopy and dual-beam-focused ion beam precision. Lattice distortion in Ni-Li regions was identified as a nucleation site for cracks. It was concluded that strain differences between inactive phases (Ni-Li) and active phases (layered oxides) during delithiation/lithiation cause a severe stress state, leading to this type of cracks.

Although cracking during charge/discharge cycles caused by anisotropy-induced stress states accounts for a degradation mechanism of major concern, other reasons may account for the short cycle life of Ni-rich cathodes. L. Zhu et al. found that approximately one-third of commercial NMC811 cathode material had some sort of crack at the micrometer scale [8], occurring solely due to the fabrication process, co-precipitation. Even though the exact percentage of defects is highly influenced by the specific manufacturer, the result obtained indicates that the manufacturing process possesses a significant role in the structural integrity of NMC cathodes. Furthermore, CR2032 coin-type cells with graphite as counter electrode were cycled, and through X-ray nano-CT imaging there were found several cracks throughout (binded) active particle sheets and delamination between those and the current collector, especially for samples cycled to higher charging cut-off voltages. Since these phenomena are associated with the macro-scale, it is suggested [8] that the carbon binder may be the source of this mechanical degradation.

Another mechanism responsible for the performance fading of Ni-rich cathodes, already mentioned for LiNiO${}_{2}$, is Li/Ni${}^{2+}$ cation mixing. During delithiation/lithiation, Ni${}^{2+}$ ions migrate to Li layers, blocking Li${}^{+}$ diffusion channels [7,66], ultimately decreasing cycle life [68]. Ionic conductivity is also hampered by Ni${}^{4+}$ ions forming at highly deintercalated states, which causes side reactions between electrode and electrolyte that result in transformations from active Ni phases into inactive NiO rock-salt phase.

NMC and NCA electrodes have different susceptibility to each degradation mechanism affecting Ni-rich cathodes. Li et al. performed a thorough evaluation on the long-term cyclability of both layered oxide cathodes [69], in an effort dedicated to comparing the effect of different metallic ions, Mn and Al, in Ni-Co oxide cathodes. The authors highlight that bare NMC cathodes with high Ni content suffer from accelerated degradation compared with NCA cathodes. The main degradation mechanisms considered were the following:Electrolyte oxidation is more severe for NMC cathodes, owing to both increased rate of Mn dissolution, opposed to Al dissociation from NCA and higher susceptibility for irreversible phase transitions, mainly from layered oxide to NiO rock-salt phase. Even though this transformation occurs for both types of cathodes, the tendency of Mn-based layers to form a spinel-like structure promotes the formation of this phase.Particle pulverization, or separation of primary particles, is caused by severe volume changes during cycling. The transition from H2 to H3 and vice-versa introduces severe anisotropic lattice changes, which serve as a driving force for intergranular cracks. NCA suffers larger volume changes than NMC cathodes, suggesting that Mn is more effective than Al at mitigating volume changes.The migration of cathode dissolution to graphitic anodes produces damages at the solid electrolyte interphase on the anode side, leading to a capacity decay. Again, the couple NMC/graphite seems to be more susceptible to this degradation mechanism in comparison with NCA/graphite. It was also concluded [69] that dissolution/crossover of transition metals and irreversible phase transformation in NMC outweighs the susceptibility to particle pulverization of NCA, leading to a superior capacity decay of the former type of electrode. Reducing the chemical activity of the electrolyte-electrode interface is a key factor towards achieving enhanced cyclability of NMC-based LIBs. On the other hand, intergranular cracks (particle pulverization) of both Ni-rich cathodes severely affect the structural integrity, and, thus, the cycle life of the cell.

### 2.8. Solutions for Mitigating Low Ionic Conductivity and Capacity Fading of Ni-Rich Cathodes

Most solutions proposed for improving the electrochemical performance of Ni-rich cathodes involve either doping the electrode, surface coating, controlling particle size, crystalline structure, porosity, and/or film thickness [5,6,53,54,55,56,59,60,70,71].

From these alternatives, surface coating and doping of the inner surface have not only been the focus of research interest, but also the most mature solutions proposed.

In practice, surface coating protects the surface of the cathodes from exposure to the electrolyte [65]. This significantly slows the kinetics of side reactions (electrode/electrolyte) and irreversible phase transformations (Ni${}^{4+}$/electrolyte) [66]. Additionally, some coatings reduce electrode/electrolyte interface resistance, thus increasing the conductivity of bare cathodes [64]. On the other hand, doping elements decrease the vulnerability to particle pulverization [7].

Surface Coating 

The outer surface coating can be achieved either by depositing a nanometer coating film on the surface of a bare cathode or by creating composite layered materials [64]. Regarding coating film design, Zhu et al. provide an extensive comparison between several coating strategies for NMC111 cathodes [62]. Among several experimental works cited ZrO${}_{2}$, MgO, TiO${}_{2}$, and, particularly, Al-based coatings stood out as remarkable options for increasing structural stability. On the other hand, carbon coating leads to significant improvements in ionic and electron conductivity, besides also improving structural stability. Chen et al. [54] synthesized carbon-coated NMC811 cathodes using PVDF as a binder and tested several samples with different wt% in PVDF at 0.2C. Both initial discharge capacity (∼190–195 mAh·g${}^{-1}$) and Coulombic efficiency were essentially unaffected by carbon content, while the capacity decay after 100 cycles improved from 6.85% to 1.26% for 2.5% PVDF content. Fluoride coating also improves chemical stability by inhibiting side reactions, provided F-based electrolytes are used. It is clear that for any coating layer to be effective, a compromise between conductivity and chemical stability needs to occur; thin layers may optimize the conductivity of coated cathodes, but a minimum thickness is required for preventing direct contact with the electrolyte [64].

Another important feature of surface coating is its potential for increasing thermal stability and safety of NMC-based cells. Wu et al. [72] have synthesized a Bismaleimide/trithiocyanuric acid oligomer for coating NMC532 electrodes and studied decomposition at high temperatures. The authors cycled CR2032 coin-type cells and obtained similar capacity retention and rate performance, compared with non-coated cathodes. However, decomposition temperature for the former was much higher (317 vs. 284 ${}^{\circ }$C) and lower heat generation was measured (599 vs. 824 Jg${}^{-1}$). In this experiment, a 1M LiPF${}_{6}$ electrolyte was used dissolved in equal parts of ethylene carbonate and diethylene carbonate. Both experimental studies are further characterized in Figure 10, allowing for further comparisons between both coating alternatives.

Composite cathodes 

Usually, composite cathodes consist of combining different oxides in a layer-wise structure (e.g., xLi${}_{2}$MnO${}_{3}$(1-x)NMC, Li${}_{2}$MnO${}_{3}$NMC, LiFe${}_{0.4}$Mn${}_{0.6}$PO${}_{4}$, LMO-NMC, LCO-NMC) or an inactive structured layer (e.g., nanostructured carbon, Al(OH)${}_{3}$). In the same manner, as combining different transition metal ions in the same layered structure has led to successful compromise solutions between thermal, structural, and cycling behavior of Ni-rich cathodes, coating NMC electrodes with LiFePO${}_{4}$ could potentially result in combining the high capacity and high cycle life of the respective cathode types. Zhong et al. [53] have reported a nano LiFePO${}_{4}$ coated LiNi${}_{0.82}$Co${}_{0.12}$Mn${}_{0.06}$O${}_{2}$ composite prepared via mechanical fusion. Since both material systems are electrochemically active, a higher specific capacity may be achieved. However, it is necessary to mix LFP with another conductive coating, carbon, given that the former is a non-conductive material. NMC, LFP, and the NMC@LFP composite CR2032 coin-type cells were tested in the voltage range of 3.0–4.2 V. The composite cathode exhibited a similar discharge plateau as LFP at 3.40 V, which suggests that the coating layer significantly inhibits side reactions occurring with the electrolyte: 1M LiPF${}_{6}$ dissolved in ethylene carbonate/ethyl methyl carbonate/dimethyl carbonate (3:2:5 vol.).

Cylindrical 18,650 full batteries with the composite cathode were also tested. At 1C, discharge capacities obtained for NMC/NMC@LFP cathodes were as follows: initial discharge capacity (mAh·g${}^{-1}$) of 184.9/180.3, discharge capacity after the 500th cycle equal to 130.0/165.3, corresponding to 70.3%/91.7% capacity retention (70.3% and 89.4% considering the initial capacity of NMC).

NMC surpasses NMC@LFP in terms of rate performance, providing 182 mAh·g${}^{-1}$ versus 176.5 mAh·g${}^{-1}$ discharge capacity at 8 C. Obviously, this result was to be expected, owing to the poorer rate performance of LFP cathodes. Nevertheless, it is only a slight decrease, especially when compared with the improvement obtained in cycle life.

An accelerating rate calorimeter was used to test the thermal behavior of NMC@LFP and NMC 18,650 batteries. Temperature-time curves were analyzed based on a division in three zones: (**1**) an approximately linear behavior is observed corresponding to heating before the onset of SEI decomposition. Both types of cathode exhibit similar behavior and the onset temperature calculated is approximately the same (91 ${}^{\circ }$C), corresponding to the safe usage temperature; (**2**) SEI decomposition occurs simultaneously with regeneration at the anode/electrolyte interphase, an exothermal reaction. For this reason, the heating rate starts to increase, until the thermal runaway temperature is achieved; (**3**) After thermal runaway triggers, the heating rate increases exponentially, and a sudden increase in temperature occurs. The NMC@LFP cathode outperformed non-coated NMC in thermal safety, achieving thermal runaway at a higher temperature (153 ${}^{\circ }$C vs. 147 ${}^{\circ }$C), and after a longer period of time (989 vs. 796 min after the onset of SEI decomposition). In Figure 11, Charge/discharge curves and Nyquist plots for NMC and composite cathodes, and thermal runaway response of full cells using both cathode types are displayed.

Doping 

Doping elements partially replace ions in Ni-rich cathodes, usually resulting in decreased Li${}^{+}$/Ni${}^{2+}$ disorder, thus improving structural stability and resistance to microcracks initiation. The advantages of doping this type of cathode are closely related to achieving superior structural stability. For instance, Nb${}^{5+}$ has been used for increasing capacity retention, while Ti${}^{4+}$ doping is associated with improvements in electrochemical properties at low temperatures. Mo-doped NMC cathodes have also been studied, and results obtained suggest that it may improve the discharge capacity [8]. In another experimental work [55], NCA cathodes with 95% of Ni content were doped with W, and delivered an initial discharge capacity of 242 mAh·g${}^{-1}$ (at 0.1C rate) and maintained 77.4% of its original capacity after 1000 cycles, while its non-doped counterpart only retained 14.5%. For the experimental work, 1.2 M LiPF${}_{6}$ dissolved in ethylene carbonate/ethyl methyl carbonate was used as electrolyte. Al-doping of NMC cathodes has also led to an abundance of research activities. Jeong et al. [57] analyzed the effect of Al doping at the lattice structure of LiNi${}_{0.80}$Mn${}_{0.05}$Co${}_{0.15}$O${}_{2}$ and found a lower level of anisotropy in shrinkage and expansion during delithiation/lithiation. Since stresses arising from this anisotropy are responsible for the driving force in crack propagation of NMC cathodes, the structural integrity was, thus, improved. After just 20 cycles, the doped specimen had a higher discharge capacity than the non-doped electrode, and after 100 cycles discharge capacity was 214.0 vs. 197.9 mAh·g${}^{-1}$. Furthermore, both cathodes were subjected to heating until achieving 600 ${}^{\circ }$C, and the Al-doped NMC transformed from its layered structure to a spinel structure, while non-doped NMC experienced a further transformation into the rock-salt phase, in an electrolyte constituted by 1.2 M LiPF${}_{6}$ dissolved in ethyl carbonate/dimethyl carbonate (1:1) solvent.

A study on Al-doping of NCA cathodes was conducted by Zhou, Xie, and Li et al. [58] to understand the beneficial and detrimental effects associated with increasing Al content. Initial Coulombic Efficiency (ICE) and initial discharge capacity decreased with increasing dopant content, which was attributed to the formation of LiAlO${}_{2}$ and Li${}_{5}$AlO${}_{4}$ (reduced active Li in the cathode). Al is electrically inactive, and excessive doping hampers the specific capacity of cathodes, which suggests there should be a maximum dopant value. After testing several pouch full-cell samples, at 50 ${}^{\circ }$C, with graphite as the counter electrode, and dissolved LiPF${}_{6}$ as the electrolyte, they discovered that further increasing the Al content above 5.6% did not significantly increase capacity retention. For 5.6% (LiNi${}_{0.85}$Co${}_{0.094}$Al${}_{0.056}$O${}_{2}$) the capacity retention was ∼88%, while for 2.3% and no Al doping the capacity retention was ∼55% and 53%, respectively. After 100 cycles, the 5.6% cell already had the highest capacity, 157 mAh·g${}^{-1}$. Furthermore, structural analysis on cathodes after 100 cycles was conducted to relate capacity retention with the morphology of each electrode. While the LiNi${}_{0.85}$Co${}_{0.15}$O${}_{2}$ cathode exhibited severe cracking in secondary particles, Al-doped samples showed far less cracking at the microscopic level, and no significant differences between cathodes with 5.6% and higher Al content were observed. In Figure 12, Figure 13 and Figure 14 the performance of the W-doped NCA, Al-doped NMC, and Al-doped NCA cathodes, respectively, is presented. An increase in rate capacity is observed for all specimens.

Considerations Regarding the Combined Effect of Inner Doping and Surface Coating of Ni-Rich Cathodes

All the examples that have been shown thus far refer to either doping or coating of these types of cathode. However, both may be combined to, simultaneously, improve mechanical integrity and chemical stability, while increasing conductivity. For instance, Tang et al. [56] used La${}_{2}$Zr${}_{2}$O${}_{7}$ coating and Zr doping for simultaneously increasing ionic conductivity and structural stability of LiNi${}_{0.8}$Mn${}_{0.1}$Co${}_{0.1}$O${}_{2}$. Assembling CR2025 coin-type cells at 0.1C, different w% of La${}_{2}$Zr${}_{2}$O${}_{7}$ (1%, 2%, and 3%) were tested and compared to bare NMC and Zr-coated NMC cells. Comparing with the pristine cell, cells with 2% of coating achieved an initial discharge capacity of 191.4 mAh·g${}^{-1}$(vs. 195.9 mAh·g${}^{-1}$), and after 100 cycles had a discharge capacity of 177.5 mAh·g${}^{-1}$ (vs. 117.1 mAh·g${}^{-1}$), or in terms of capacity retention, 92.7% vs. 59.8% (90.61% vs. 59.8% in terms of initial discharge capacity of pristine NMC). For Zr-doped (non-coated) NMC cells, the capacity retention was equal to 86.3% (80% comparing to pristine cells). Furthermore, once again the rate performance of coated cathodes was significantly improved, which is clearly observed in Figure 15. At 0.5C, their initial discharge capacity was greater than for pristine cells, and after 500 cycles cells with 2% of coating still retained 118.8 mAh·g${}^{-1}$, while after just 100 cycles pristine cells had a value lower than 100 mAh·g${}^{-1}$. Another important characteristic of these cells is their one-step synthesis, consisting of calcination of a gel mixture containing La(NO${}_{3}$)${}_{3}$6H${}_{2}$O, Zr(NO${}_{3}$)${}_{4}$5H${}_{2}$O, and LiNi${}_{0.8}$Mn${}_{0.1}$Co${}_{0.1}$O${}_{2\;}$ under an oxygen atmosphere. Coating and doping need to be achieved via processes facilitating industrial application and envisioning low-cost solutions, otherwise, their applicability will be impeded. Even though surface coating prevents undesired side reactions and phase transformation, thus improving cycle life, if accomplished with multiple additional manufacturing steps, it significantly increases processing costs [7]. Integration in the original manufacturing process and single-step coating are ideal solutions. For instance, G. Chen et al. [54] opted for PVDF/NMP as a binder for carbon-coated NMC cathodes due to its usage in electrode manufacturing at the industrial level. The previous examples show how a cost-effective design of coating and doping may be achieved in next-generation LIBs with Ni-rich cathodes.


**Density and Particle Size Effects on Electrochemical Performance**


Theoretical specific capacity and energy, at the cell level, are enhanced by decreasing the level of porosity in the cathode. However, the practical values obtained are a function of the microstructure and excessive compaction leads to a degradation in electrochemical performance. This is mainly attributed to the transport mechanism that leads to ionic and electronic conduction [70]. While the former is favored by high porosity, due to the formation of channels for ionic transport, the latter benefits from increased contact between particles. Clearly, a trade-off between maximum compaction (high theoretical energy density and electron conductivity) and sufficient porosity (towards achieving high ionic conduction) is paramount for developing improved Ni-rich cathodes.

Schmidt et al. [70] assembled NMC111 cathodes, using 1 M LiPF${}_{6}$ dissolved in 3:7 ethylene carbonate/ethyl methyl carbonate as the electrolyte, via co-precipitation, and obtained several porosity values in the range of 18–50% by changing compaction pressure, as plotted in Figure 16. It was observed that at a very low C-rate (0.05C), low porosity cathodes exhibit the same behavior in terms of discharge capacity. However, as the rate increases to 2C, there is a significant drop in capacity for low porosity cathodes. Cathodes with 18% and 20% porosities lost most of their capacity; while for 35% porosity, only one-third of the capacity was lost, suggesting that the ideal porosity for practical applications, considering fast charging conditions, should be in the range of 30–35% for NMC111.

Hu et al. studied the evolution of rate performance in LiNi${}_{0.70}$Mn${}_{0.22}$Co${}_{0.08}$O${}_{2}$ and found that thicker electrodes exhibited lower electron and/or Li${}^{+}$ diffusion, leading to poorer rate performances [5]. This is especially true for highly porous and electrolyte fully flooded cathodes, for which polarization negatively affects electron diffusion. In dense electrodes, Li${}^{+}$ flow channels are interrupted. Given the opposing nature of the effects, after testing CR2032 coin cells at 2C, an optimum value for porosity was established at 40% to obtain maximum capacity retention. The filling electrolyte used consisted of 1 M LiPF${}_{6}$ dissolved in ethyl/ethyl methyl carbonate.

Controlling secondary and primary particle size allows controlling the final porosity obtained. Kim et al. [59] prepared, via co-precipitation, a LiNi${}_{0.91}$Mn${}_{0.03}$Co${}_{0.06}$O${}_{2}$ layered oxide with a secondary particle size of 9 m, and primary particles of size in the range of 300–500 nm. Half-cells were filled with 1M LiPF${}_{6}$ dissolved in ethylene/dimethyl/ethyl methyl carbonate electrolyte. It was due to low particle size that a good rate performance was obtained for a highly porous electrode with an extremely high value of Ni content, promoted by enhanced electrode-electrolyte contact. Increasing the current rate from 0.5C to 1C, 2C, and 5C after 5 cycles between consecutive values, discharge capacities of 208.3, 191.2, 178.4, and 70.5 mAh·g${}^{-1}$ were obtained. After dropping the current rate back to 0.5C, a discharge capacity of 183.4 mAh·g${}^{-1}$ was still obtained, a value remarkably close to the initial charge capacity at 0.5 C, as can be shown in Figure 17.

In terms of structural integrity, modifications on the crystalline structure of Ni-rich cathodes have been investigated as a solution for mitigating cracks emanating from anisotropy during volume changes. From a crystallographic point of view, nucleation sites and driving force for intergranular cracks to propagate arise from the high number of grain boundaries formed by randomly oriented primary particles structure, together with anisotropic contraction and expansion of NMC cathodes. Single crystal solutions eliminate these grain boundaries, thus allowing for volume changes to occur without intergranular crack formation [60]. Moreover, for the same material content, the electrolyte/electrode interphase area is smaller, thus inhibiting side reactions.

Fan et al. have developed a single-crystal LiNi${}_{0.83}$Mn${}_{0.3}$Co${}_{0.2}$O${}_{2}$ cathode material system (SC-NMC) via co-precipitation preparation [60]. To conduct electrochemical measurements and compare results with commercial NMC cathodes, CR2032 coin-type cells and pouch cells were fabricated and cycled between 2.75 and 4.4 V at 25 ${}^{\circ }$C, and 1 M LiPF${}_{6}$ in ethyl carbonate/diethylene carbonate (EC/DEC, 1:1 in volume) was chosen as the electrolyte. Besides achieving a higher initial discharge capacity, the capacity decay after 150 cycles was less than half of commercial NMC. Increasing the temperature to 55 ${}^{\circ }$C, the difference in capacity retention was further enhanced. The energy density (with respect to the mass of active material) after cycling was also significantly increased, particularly at a higher temperature, as can be observed in Figure 18a. To test both electrodes in real-life EV battery conditions, both cathodes were paired with a SiO/C composite anode in pouch cells at 1C and 45 ${}^{\circ }$C. While similar initial discharge capacities were obtained (188.2 mAh·g${}^{-1}$ vs. 184.1 mAh·g${}^{-1}$ for single crystal), the capacity retention and specific energy after 500 cycles were far superior for the latter (43% and 150 Wh·kg${}^{-1}$ for commercial NMC; 84.8% and 225 Wh·kg${}^{-1}$ for single crystal). However, the rate performance obtained for the synthesized cathode was inferior to the one measured for multi-crystal NMC. The same tendency was observed for LiNi${}_{0.5}$Mn${}_{0.3}$Co${}_{0.2}$O${}_{2}$[71], for which a multi-crystal structure outperformed a single-crystal at a high current rate (5 C)-identified as M- and S-NCM in Figure 18b-in studies conducted using 1 M LiPF${}_{6}$ dissolved in a mixture of ethylene carbonate/dimethyl carbonate solvents (1:1 vol.).

## 3. The Negative Electrode (Anode)

Virtually every commercially available battery for mobile applications has batteries containing a graphitic anode [73], which remains to be the most compatible with Li-ion cathodes. Julien et al. [74] thoroughly described the requirements for an ideal anode. Limiting ourselves to some of the most relevant features, the abundance of graphite anodes owes to its:Low chemical potential μ(Li)-μ(LiC${}_{6}$) ≈ 0.1 eV, meaning that a lithiated graphite (LiC${}_{6}$) anode shows an equilibrium plateau discharge voltage that is 0.1 V lower than a Li-metal anode, for a cell containing a similar cathode and internal resistance.Significant worldwide reserves.Good electrochemical stability [75].Safer than Lithium in case of fire.

Anode technology has been reasonably stable throughout the years, but the continuous development of Li-ion cathodes, new materials, coatings, and manufacturing processes pushes for new studies. Since anode degradation is accountable for much of full cell degradation [76,77], improvements at the anode component level are important for developing long-lasting batteries for mobile applications [77]. Presently, Carbon (usually Carbon Black) coated graphite, Li-metal, and Silicon-based anodes are widely regarded as the major alternatives to replace graphite as the most important anode for LIBs. In fact, Si is already incorporated in some anodes as a coating substance for graphitic material systems.

The main anode types currently in use for mobile applications are summarized in Table 4. Additionally, the main properties of concern are exhibited. Graphite possesses the lowest voltage (V vs. Li${}^{0}$/Li${}^{+}$) among the alternatives (Lithium-Titanium oxide Li${}_{4}$Ti${}_{5}$O${}_{12}$ (LTO), and Silicon/Silicon Nano-Wire (SiNW)). However, graphite anodes not only possess relatively low specific capacity (theoretically, 372 mA·g${}^{-1}$) but also have a typical cycle life of the same order as standard NMC cathodes, which means that graphite can limit the cycle life of the cell [78]. In fact, in a recent study conducted by Dose et al. [79], it was found that the graphitic anode may be responsible for triggering or accelerating the degradation of Ni-rich NMC/Graphite full cells. It is suggested that anode slippage (capacity fading due to continuous Li loss, consumed by the formation of a graphite solid electrolyte interphase (SEI)) leads to progressive capacity loss. Moreover, after just 310 cycles, the initial LiC${}_{6}$ phase formed on the surface of lithiated graphite anodes is lost. Instead, an XRD scan reveals the presence of a LiC${}_{12}$ phase. As a direct consequence, the low voltage plateau of 73 mV (versus Li/Li${}^{+}$) is no longer achieved, and a new higher plateau, at 110 mV (versus Li/Li${}^{+}$) is obtained, leading to a higher cutoff potential of the cathode. The authors [79] also identify this rise in cutoff potential as responsible for numerous NMC cathodes degradation mechanisms - already mentioned in Section 2.7-such as higher degrees of capacity loss, transition metal dissolution, a transition from layered to rock-salt phase, and particle cracking. Dose et al. [79] also provide two strategies for prolonging cell life: (1) inhibit anode slippage, accomplished by using lower surface area graphite and/or adding electrolyte additives, in order to stabilize the SEI; (2) dynamically adjust the full-cell voltage window according to the cutoff potential increase. Evidently, a significant decrease in cycle life may be largely attributed to the chemical instability that occurs at the electrode/electrolyte interface [80]. This fact, besides motivating research to find new materials, led to the development of carbon-coated graphitic anodes. The advantages thus obtained consist of:a thinner SEI, potentially leading to higher capacity [74], as the SEI consumes Li${}^{+}$–it is a mixture of insulating compounds mainly containing lithium;a reduction of chemical instability between electrode and electrolyte leading to a great improvement in cycling performance [81].

It is highlighted that the SEI is formed spontaneously when the anode and electrolyte become in contact during charge to align their chemical potentials and the Lower Unoccupied Molecular Orbital (LUMO) of the liquid electrolyte at lower energy than the chemical potential of LiC${}_{6}$, μ(LiC${}_{6}$), leading to an electron current leak from the anode to the electrolyte. Those leaked electrons reduce Li${}^{+}$-ions and subsequently LiF, Li${}_{2}$O, and Li${}_{2}$CO${}_{3}$ are formed (if the electrolyte is the most common mixture of a carbonate solvent and LiPF${}_{6}$). The formation of these insulators inhibits the leakage of electrons to the electrolyte enabling the conduction of electrons throughout the external circuit.

Other types of Carbon-based anodes with higher capacities have been the subject of research efforts, namely Carbon nanotubes and graphene. However, their use is limited due to the cost of the manufacturing process and discharge capacity degradation, respectively, although the latter has shown promising recent results, suggesting that this barrier may be overcome [82].

Compared with the typical values shown for graphite anodes the LTO and SiNW anodes show a significantly higher cycle life as well as specific capacity, which clearly explains why these anodes are viewed as the successors for graphite anodes. Both anode types have reached a technological stage that allowed their implementation on LIBs, as shown in Table 4. Nevertheless, LTO presents itself as a more mature technology, as it is commercially available [83,84,85].

According to Toshiba, their LTO based batteries are extremely safe, with little risk of thermal runaway [85]. Similar information is provided by other sources [74,85] stating that LFP/LTO cells perform especially well on safety tests. Besides safety reasons, this anode outperforms graphite on cycle life [86]. As such, it arises as an ideal technology for applications requiring a high number of cycles, such as industrial or military applications with no restrictions on total battery weight (thus, specific energy), or public transportation vehicles.

### 3.1. Silicon Anodes

Silicon anodes have long been the subject of intensive research because they are relatively inexpensive (2nd most abundant element on the surface of Earth), very high specific capacity (theoretical capacity approximately 4200 mAh·g${}^{-1}$ for Li${}_{22}$Si${}_{5}$ and 3579 mAh·g${}^{-1}$ for Li${}_{15}$Si${}_{4}$ phases) [42,43] and working potential, at around 0.4 V–superior to commercial graphite but lower than LTO [87]. Depending on the full battery system, the fact that its operating voltage is higher than for a graphitic anode may be an advantage since it generally leads to safer Charge/discharge cycles. However, due to its low density, high volumetric expansion associated with cycling (lithiation), electrical conductivity, and unstable SEI film formation [88], half- and full-cells using this anode type have suffered from performance degradation at an early stage [89]. The most promising solutions for overcoming this problem are nanocrystallization [90] and the development of Si-composite materials (frequently combined). From several pure-Si nanomaterials currently being studied, SiNW stands out as a successful implementation, with some applications at the battery level, such as Amprius Technologies battery developed for Airbus Zephyr S Pseudosatellite [77,91]. The main challenge that prevents its industrial development is high manufacturing cost and/or severe degradation mechanism. The former is related to the complexity of design solutions proposed for mitigating the latter, which is mainly caused by the disadvantages mentioned above. The tremendous volume variations during Charge/discharge lead to a direct capacity loss (severe cracking), and broken/renovated SEI films, which consumes Li ions and the electrolyte, hampering ionic conductivity and reversibility (the initial formation of an SEI layer is responsible for very low Initial Coulombic efficiency [92]); similarly to Ni-rich cathodes, mechanical stresses induced by volumetric expansion/contraction lead to severe crack initiation/propagation, and particle pulverization [87]. The advantages of using nanoparticles in composite materials are closely related to increasing ionic conductivity and designing porous structures capable of accommodating large volume changes.

### 3.2. Importance of Particle Size for Si Nanoparticles (SiNPs) Anodes

Contrary to graphite anodes, for which Li intercalation occurs, Si anodes rely on a chemical bond between Si and Li [93] and this chemical principle is the main responsible for the critical fracture of Si particles during lithiation (accompanied by large strains caused by volumetric expansion). Controlling particle size is paramount for preventing this type of fracture to occur. Earlier research work on the topic modeled the behavior of Si particles during cycling as a single-phase material subjected to diffusion-induced traction hoop-stresses at the center, where it was believed that fracture first occurred [94]. However, Liu et al. [93] conducted in situ TEM observations and found that the fracture was initiated at the surface of nano-Si particles. A Finite Element Analysis modeling the lithiation process, based on a two-phase model, with a Si-core and Li${}_{x}$Si shell, was conducted and a completely different stress-state was found. The results highlighted a compression at the center of the particle and a tensile state on the outer layers. This arises from the mass gradients during lithiation with decreasing Li${}_{x}$Si concentration towards the center. This was a disruptive approach that contradicted previous models, predicting traction at the center and compressive hoop stresses across the particle radius. The results were compliant with the crack initiation phenomena at the surface, with a subsequent propagation dependent on particle size. As particle size increases, so do the released elastic strain energy and, consequently, the crack driving force. A critical value of ∼150 nm was determined, below which no significant fracture was observed.

In parallel, an experimental study on the kinetics of SiNP lithiation was conducted by McDowell et al. [95]. It was found that using extremely small particles negatively affects the delithiation/lithiation, slowing down the process. A particle size of 20 nm was found as ideal for increasing lithiation speed. Both works combined provided the upper and lower limit bounds for nano-Si particle size, based on sound interpretations of the physical phenomena here involved. A similar result was obtained in a different experimental work [96]. Nano-Si anodes of different particle sizes were cycled with increasing current rate, from 0.1 to 10 C. A large capacity fade was observed for Si electrodes with crystallite size equal to 30.9 nm, and results for smaller sizes suggest the same tendency. These competing effects indicate that crystallite size may be tailored and optimized for each specific application, depending on the relative importance between rate performance (larger particle size) and cycle life (smaller particle size).

### 3.3. Design of Nano-Si/Carbon Composite Anodes

Given the disadvantages of pure Si anodes, a successful design for a composite Si anode must target an increase in electrical conductivity, density, and chemical stability, besides being able to alleviate the mechanical stress induced by volume expansion. Since studies on composite Si anodes as a replacement to graphitic electrodes began, numerous designs combining nano-Si or SiO${}_{x\;}$as active materials, with an outer matrix, most frequently carbon, have been proposed [96,97,98,99,100,101,102,103,104].

As the mature technology in LIBs anodes, graphite presents several advantages enabling their use in anodes. As a matrix for Si/C composites, most of those advantages may be retrieved; graphite as a structure for SiNP increases electrical conductivity and density of the powder after being tapped, enhances chemical stability, and possesses an ideal shape for alleviating volume expansions [64]. Furthermore, it is a natural lubricant, providing calendaring compatibility, favorable for attaining high practical energy density [88]. On the other hand, graphene has a high specific surface area, high electrical conductivity, and large mechanical strength [105]. However, there are two common issues associated with carbon coating: (1) cracking and (2) mass deficiencies of the carbon layer allowing the electrolyte to contact with SiNP, thereby incurring the repetitive growth of an SEI layer [106].

In 2018, Li et al. [64] published a review on the most relevant approaches for developing industry-ready Si composite anodes. At the time, the tendency was to synthesize anodes using SiNP dispersed in SiO${}_{x}$ as the active material. Structural and chemical stability are, thus, favored to the detriment of theoretical specific discharge capacity, which is still much higher than for common graphitic anodes. While some research groups were focused on increasing capacity retention [97,101,102,104], others studied different approaches for reducing initial irreversible Li${}^{+}$ loss due to SEI formation [98,99]. Also, the structure of composite materials consisted of Si-based shells–core/shell or yolk/shell designs. The outer shell (1) acts as a buffering layer, mitigating volume expansion; (2) blocks direct contact between Si and electrolyte, reducing chemical side reactions between them; (3) if composed of functional materials (e.g., carbon), electronic and ionic conductivity may be improved. The yolk/shell configuration consists of incorporating a void space between Si and carbon surfaces, providing a solution for nanostructuring Si, the formation of a stable SEI, well-controlled pore space, and scalable fabrication [104]. The clear mechanical advantage of this structure, compared to core/shell, is allowing for volume expansion through void-filling, thus increasing capacity retention and SEI stability [104].

### 3.4. Progress in Si Composite Anodes Design

In recent years, research on composite Si anodes has mainly followed the core/shell and yolk/shell approach towards designing electrodes with superior capacity retention and/or rate performance [88,105,106,107,108,109,110,111,112,113,114]. In Figure 19, it is possible to identify both design strategies. While the former consist of a dual layer design with a contact between the inner core and outer shell, the latter incorporates a void space between two substances thus theoretically allowing for better accommodation of volume expansions, in detriment of electrical and ionic conductivity. Besides Si/carbon, multiple multi-material solutions for the matrix have been proposed, often combining carbon with a metallic material. Throughout this subsection, several experimental studies conducted in recent years dealing with Si composite anodes will be described. To facilitate a comparison between results, electrochemical performances are summarized in Table 5.

Zhang et al. [111] developed a multilayer carbon matrix, graphene oxide, toughened by cross-linked carbon nanotube (CNT) chains surrounding SiNP (Si@C-CNT-Cu). Cu conductive particles were deposited throughout each layer. The combination of metallic particles with a carbon shell favors high electrical conductivity. CNTs aid the carbon matrix in its mechanical function of accommodating volume expansion of SiNP, besides improving thermal stability and electronic conductivity [74]. Benefits in using 3D porous structure (multilayer) instead of 2D are as follows: improved ionic and electronic transport pathway; enhanced electrolyte infiltration; better accommodation of volume changes. They also identify electrical conductivity as the key issue for improving 3D structures towards superior Si anode performance, which explains the importance of adding Cu particles. Furthermore, a precursor gel composed of Si nanoparticles, CNTs, polyvinylpyrrolidone and CuO was fabricated, and glucolactone was further added for releasing Cu${}^{2+}$. Finally, carbonization was achieved by heat treatment. Regarding the synthesis process, it is suggested that in situ techniques enable a more homogeneous deposition, while also preventing oxidation reactions. Half-cells were prepared with 1M LiPF${}_{6}$ dissolved in ethylene carbonate/dimethyl carbonate (1:1 volume ratio). The results obtained are presented in Figure 20a. The anode showed good capacity retention and a remarkable rate performance, as specific capacity returned to its original value after testing at several rates. More recently, Liu et al. [108] incorporated a Cu network in a Si/carbon shell composite anode to increase the mechanical performance and the electrical conductivity of the porous structure. Cu was chosen due to its chemical stability with electrolytes and superior electrical conductivity. Furthermore, Cu is a ductile metal, owing to its atomic arrangement, with a face-centered cubic crystal structure. Ductility is of utmost importance for increasing the structural integrity of the composite anode since it increases the amount of dissipated energy during fracture, one of the major mechanisms for improving the crack resistance of materials. To enhance the beneficial effect of incorporating Cu, a thin metallic layer wrapping Si was obtained, as well as an outer layer of carbon. Pyrrole and carbon nitrate were used as carbon and copper precursors, respectively. A dry mixture of nano-Si particles and both materials were heated at 900 ${}^{\circ }$C in an Ar-rich atmosphere, leading to carbonization by carbothermic reduction. The composite Cu/Si anode, named C-SCP, was compared against Si/C and Si NP control samples, and specific capacities were measured with respect to active Si mass. C-SCP outperformed the remaining electrodes in every electrical measurement, performed in half-cells using 1M LiPF${}_{6}$ dissolved in ethylene carbonate/diethylene carbonate (1:1 vol.) with 25% of fluoroethylene carbonate as an additive. Chen et al. [105] also studied the advantages of incorporating CNTs, using them for encapsulating SiNP coated with carbon, wrapped around graphene sheets (Si@C/CNTs@GS), thus achieving a hierarchical protective structure. Flake graphite (GO) and NSi-OH were used as carbon and SiNP precursors, respectively. Si@C/CNTs was obtained by mixing with CNTs dispersed in water. This mixture also led to a carbon coating of Si via further spray drying. The final composite was obtained via spray drying followed by calcining. Tests using CR2430 button cells were performed for pure SiNP, Si@C/CNT, and Si@C/CNTs@GS anodes, using 1M LiPF${}_{6}$ solution in a 1:1:1 ethylene carbonate, diethyl carbonate, and ethyl methyl carbonate as electrolyte. The graphene composite showed the smallest initial Charge/discharge capacity, but superior cycling stability (more than double of discharge capacity compared to the single protective function anode, after 130 cycles). Moreover, CE was approximately 100% after just the first cycle. The authors explained the superior cycling stability with a lower volume expansion of CNTs allied with high strength of (wrinkled) graphene, also leading to an improvement in rate capability, evidenced in Figure 20b. Cai et al. [110] were able to synthesize a SiO composite by embedding the active particles in a TiO${}_{2}$ coating shell and 3D Carbon Nanofibre (CNF) web. The metallic coating enables extra Li${}^{+}$ reaction sites and additional protection against volume expansion (due to its high strength nature). The composite was prepared by sol-gel coating and electrospinning, with PAN as the carbon precursor. Electrochemical tests performed in coin-type half cells compared SiO, Si/CNF, and Si@TiO${}_{2}$/CNF. ICE and capacity retention for both Si/CNF and Si@TiO${}_{2}$/CNF were higher than for SiO, an improvement the authors attributed to the protection against reactions of highly active Si particles with the electrolyte, 1M LiPF${}_{6}$ dissolved in mixed ethylene carbonate and dimethyl carbonate (1:1 vol.). For SiO@TiO${}_{2}$/CNF, and after going through different values of current density, capacity retention at 0.2 mA·g${}^{-1}$ was 81.5%. The real benefits of the TiO${}_{2\;}$coating shell are a superior initial/first cycles discharge capacity and improved rate performance, especially at moderate current densities, as can be seen in Figure 20c.

Hu et al. [109] recognized the superior electrochemical performance achieved by Cui et al. with their successful yolk-shell design [104], further improved by Zhao et al. [115], but the authors suggested that having a single layer of carbon shell is insufficient for preventing the structural fragility of Si/C anodes. To further increase the structural stability of a yolk-shell Si/C composite, they developed a composite anode with a double carbon shell, which also increases conductivity, separated by layers of SiO${}_{2}$ (Si@SiO${}_{2}$@C). Particle sizes used for synthesizing the composite were as follows: (1) diameter of 50–100 nm for SiNP (respecting the critical size of 150 nm, as stated by Liu et al. [93]); (2) for the SiO${}_{2}$ layer, 2 nm. RF resin was used for coating a carbon layer of about 20 nm, and another layer of SiO${}_{2}$ of equal size was formed. The final carbon layer, the outer shell of the composite, was obtained by polymer dopamine (PDA) coating and further heat treatment on an N${}_{2}$ rich atmosphere at 900 ${}^{\circ }$C, converting PDA into carbon. The final step consists of etching, a chemical process here employed for removing O${}_{2}$ layers, and HF aqueous solution was used as the etchant. Etching for prolonged periods of time also has the effect of dividing Si nanoparticles into several particles. Single carbon shell and Si@SiO${}_{2}$@C composites were tested in coin-type half cells, and the electrolyte used consists in 1M LiPF${}_{6}$ in a 1:1 *w*:*w* mixture of ethylene carbonate and dimethyl carbonate, with an amount of electrolyte in a cell of about 100 L. Initial discharge capacity obtained for the former (648 mAh·g${}^{-1}$) was smaller than the discharge capacity of the latter after 200 cycles. Moreover, the double carbon layer anode not only has a much higher cycling performance but also displays superior rate capability (for instance, the latter shows ∼340 mAh·g${}^{-1\;}$at 2C, and for the same current rate the discharge capacity of the single carbon shell electrode was ∼110 mAh·g${}^{-1}$). Charge and discharge capacity, rate performance, and Nyquist plots for these anodes are exhibited in Figure 21, identified as Si/C–HF (single carbon shell), Si/C–PDA (single Si core with double carbon shell), a control sample to study the effect of using several smaller Si nanoparticles, and Si/C–PDA–B (Si core broken into smaller Si nanoparticles with double carbon shell). Hu et al. [109] attributed these electrochemical improvements to (1) a better accommodation of volumetric expansion, thus improving cycle life (while also increasing structural integrity) and (2) superior SEI stability. An et al. [107] aimed at developing nano-Si composite anodes using commercially sound processes—a scalable fabrication with competitive costs—while still maintaining the electrochemical performance of a Si composite anode. A nanoporous silicon@carbon (NP-Si@C) was synthesized in a CO${}_{2}$ atmosphere, using Mg${}_{2}$Si as the precursor, thus reducing fabrication to a one-step procedure, easily scalable, while simultaneously improving the sustainability of Si anodes (CO${}_{2}$ is a more ecological carbon precursor than PAN, and others). Cells with Li foil as counter electrodes were assembled, and different carbon contents were tested. Curves obtained for different specimens are presented in Figure 22. A different electrolyte from those described so far was used—1M LiC${}_{6}$F${}_{6}$NO${}_{4}$S${}_{2}$ in cosolvent of 1, 3-dioxolane and dimethoxymethane (1:1, *v*/*v*) with 1 wt% LiNO${}_{3}$. It was observed that as the percentage of carbon increases, (1) so does ICE (lower side reactions, (2) initial specific capacity decreases, and (3) porosity increases (particularly important for structural stability). The best results in cyclability and rate performance were obtained for 22% of carbon, corresponding to an average particle size of 240 nm. The anode exhibits superior capacity retention - the highest specific capacity after 120 cycles. After increasing the rate from 0.1 to 5 A·g${}^{-1}$ (0.1, 0.5, 1, 2, 3, 4, 5 every 10 cycles) and returning to 0.1 A·g${}^{-1}$, the reversible capacity obtained was 99% of its initial value. The authors suggest this a consequence of higher conductivity (higher carbon content) and structural stability. From the results here listed, it appears that the carbon precursor used for obtaining a coating layer significantly influences the performance of composite Si anodes. Ma et al. [106] were able to study these effects by synthesizing Si@C using four different precursors: poly(vinyl alcohol)/melamine resin (Si@C${}_{MR}$), leading to yolk/shell structure, resorcinol-formaldehyde (Si@C${}_{RF}$), polydopamine (Si@C${}_{PDA}$), and glucose (Si@C${}_{GLU}$), all three of core/shell type. Electrochemical measurements of half cells for all four anodes are presented in Table 5 and Table 6 and the respective curves are included in Figure 23. All results refer to half-cells using Li foil as counter electrode and 1 M LiPF${}_{6}$ dissolved in ethylene carbonate/dimethyl carbonate (1:1 vol.) with 5% FEC additive as the electrolyte. Si@C${}_{MR}$ outperforms the other composites in rate performance and capacity retention, despite having the lowest initial discharge capacity. Its superiority is so notable that the authors center their discussion around this material’s performance. Finally, after nano-scale analyses of the morphology of every constituent, the authors attributed the superior performance of Si@C${}_{MR}$, namely SEI stability, cycling performance, and reversibility, to its relatively high disordered structure, surface area, and void space created between Si and the carbon shell. Smrekar et al. [116] proposed a combination between SiNP, Sn nanoparticles, and graphite powder (Sn${}_{x}$Si${}_{y}$C${}_{1-x-y}$) as a solution for overcoming the two main issues associated with Si anodes—volume expansion, and poor conductivity—and, even though the structure thus accomplished does not consist of an active core and buffer matrix, the graphite included still accommodates volume changes, whereby Sn increases the ionic conductivity of the anode. A study evaluating the evolution of capacity retention was conducted by altering the proportion of each material. A liquid electrolyte consisting of 1 M LiPF${}_{6}$ and 10 wt% fluoroethylene carbonate in ethylene carbonate/ethylmethyl carbonate 1:1 *v*/*v* was used. Each composite electrode was prepared by ball milling and further mixing with binder and conductive carbon, a process similar to those previously described for the preparation of NMC cathodes. As expected, the amount of Si essentially controls theoretical charge capacity. However, capacity fading was a clear problem for anodes with a higher concentration of Si. After 120 cycles, the highest discharge capacity was obtained for a 25-50-25 (Sn-Si-C) ratio, but its capacity retention (63.7%) was lower than for a 3-3-3 ratio (74.9%). Furthermore, the same ratio led to the best rate performance. The color map in Figure 24 provides a useful representation of the evolution of capacity retention with different Si/C/Sn contents. Initially, the current rate was fixed at 0.5 A·g${}^{-1}$ for 50 cycles; after cycling at 1 A·g${}^{-1}$ during 50 cycles and returning to 0.5 A·g${}^{-1}$ for 20 more cycles, the capacity retention was 85% (120th vs. 50th cycle). This study showed how Si/C composite anodes incorporating other metallic elements, fabricated by processes well developed for commercial electrodes, and exhibiting an interesting electrochemical performance, may be developed.

### 3.5. Other Alternatives for Improving Si Anodes

While developing robust composite anodes is paramount for mitigating the major issues concerning Si anodes, combining a multi-material design with studies on different binders and electrolyte composition has led to promising experimental results. Cho et al. [117] have successfully binded Si/C with a polyamide film, SiCPA fabricated by a mixture of slurries coated on a Cu current collector. Polyamic acid was used as a precursor both to carbon (carbonization) and polyamide. Si particles of 5–10 m were used. To assess the electrochemical performance, two different test specimens were synthesized, each one with a different Si mass content, 95% and 62% (SiCPA-95 and SiCPA-62). Electrochemical tests of 2032 half-cells, using 1M LiPF${}_{6}$ in a mixture of ethylene carbonate, ethyl methyl carbonate, and dimethyl carbonate (2:1:7 vol.) as the electrolyte solution with 7 wt% fluoroethylene carbonate as the additive, revealed a higher initial discharge capacity of the former (higher Si content) and higher capacity retention after 50 cycles. On the other hand, the rate performance of the latter was significantly better. Both anodes outperformed control samples without the polymeric binder, suggesting that the carbon matrix is significantly enhanced by the strong binding force of polyamide, capable of withstanding higher mechanical stresses, while still assuring a conductive network for the active material. Li et al. [118] synthesized a hard/soft trifunctional network binder (N-P-LiPN), using partially lithiated polyacrylic acid and Nafion as raw materials. The aim was to combine a hard P-LiPAA phase, which provides strong adhesion, with a soft P-LiNF chain, responsible for accommodating volume expansion, thus fabricating a Si@N-P-LiPN anode. Not only the ICE (measured in coin-type half-cells) thus achieved was unusually high for this type of anode, but also good initial discharge capacity and capacity retention were obtained. The authors believe that the impressive electrochemical performance was caused by superior mechanical properties provided by the binder. Hardness and reduced modulus (mechanical properties relevant for contact mechanics) were measured for three different binders: Si@P-LiPAA—highest modulus and hardness; Si@Si@P-LiNF—lowest values for both parameters; Si@P-LiNF—intermediate values. Different electrolyte compositions were also tested by modification of the vol% of added FEC (0, 10, and 25). There is a synergistic effect in having both chain types, harder and softer, combined in the same binder. The former provides mechanical strength to the electrode. It is the structural element of the binder. However, these types of mechanical properties are accompanied by a more brittle behavior and cracks are more likely to propagate and lead to fracture of the binder. For this reason, the softer chains, associated with P-LiNF, are necessary for accommodating volume expansion, due to their more ductile behavior and increased flexibility. Guan et al. [119] synthesized a Fe${}^{3+}$-PDA/PAA (Fe${}^{3+}$ ions, polydopamine layer, and polyacrylic acid binder via a three-step procedure—in situ polymerization of PDA on Si particles, bonding of Fe${}^{3+}$ with PDA through hydrothermal processes, leading to an amorphous layer, and mixing with PAA, obtaining a cross-linked PDA/PAA coating of nano Si particles. As with other binders, the protective layers increase mechanical strength, while the cycling stability of the active Si material is enhanced. The authors found that, even though nano Si particles still suffer from pulverization after the first delithiation/lithiation cycle, the anode can maintain its structural integrity in subsequent cycles. Furthermore, a comparison with Si@PDA/PAA anodes was made, for which the anodic peak and peak area were greatly reduced, suggesting that Fe${}^{3+}$ plays a beneficial role in the chemical interaction with the electrolyte, prepared by dissolving 1 M LiPF${}_{6}$ and 5 wt% fluoroethylene carbonate in solvent mixture: ethylene carbonate/ethyl methyl carbonate/dimethyl carbonate (1:1:1 in *v*/*v*/*v*).

Adding elements to binders that affect side reactions between anode and electrolyte is only one possible solution for improving the kinetics of SEI formation and growth. Jia et al. [120] highlighted the detrimental effect carbonates contained in electrolytes, most commonly fluoroethylene carbonate, have in terms of SEI stability, and risk of flammability. The authors fabricated an electrolyte with higher salt concentration, a technique well established for Li-metal anodes, based on a similarity between stability issues in Si and Li-metal anodes. Moreover, high concentration is localized, since high viscosity, cost, and poor wettability, are associated with higher salt concentrations. The experimental work consisted of testing the electrochemical performance of an electrolyte material containing LiFSI (concentrated salt)-a TEP/FEC/BTFE (1.2:0.13:4 by mol) material system (triethyl phosphate, fluoroethylene carbonate, and bis(2, 2, 2-trifluoroethyl) ether), named NFE-2. Together with a control sample of LiPF${}_{6}$/EC-EMC/FEC, both electrolytes were combined with a commercial Si/C composite anode in half-cells and used as the electrolyte of full cells. The initial discharge capacity of ∼1337.8 mAh·g${}^{-1}$ was not much affected by the type of electrolyte. However, after 300 cycles, the half-cell with NFE-2 achieved a capacity retention of 73.4%, while the one containing the control sample had negligible discharge capacity. Using NMC333 as the counter electrode, the capacity retention was 89.8% after 600 cycles and below 50% after 500 cycles, respectively. In another recent work, LiPO${}_{2}$ was added to LiPF${}_{6\;}$ (LiFDP) for increasing the ICE and capacity retention of nano Si (∼100 nm) anodes [92]. Half-cells using LiFDP and LiFP${}_{6\;}$(control samples) were tested. The ICE increased from 52.9% (control sample) to 70.6% (LiFDP). Furthermore, the initial discharge capacity registered was 2190 mAh·g${}^{-1}$, and capacity retention improved from 22.4% to 64.9%. All the alternatives for improving Si anodes above listed are further characterized by their respective electrochemical characterization in Figure 25.

## 4. Future Prospects

Cobalt is essentially mined at the DRC raising social and economic concerns. An effort is being made to eliminate or at least reduce cobalt from positive electrode active material. The strategy is to use an NMC composition that utilizes less Cobalt such as LiNi${}_{0.8}$Mn${}_{0.1}$Co${}_{0.1}$O${}_{2}$. The drawback relates to the stability decrease of this cathode upon cycling as well as to the decrease in capacity after the first cycle (approximately 30 mAh·g${}^{-1}$).

On the anode side, the concerns are related to eliminating dendrite growth on the graphite surface while charging too fast or above the chemical potential of the lithiated graphite, which just differs 0.1–0.2 V from that of the Li-metal. The Li dendrites seldom become short circuits leading to explosions. These complications have become critical as more and more electric scooters are in use and must be charged at home, namely in populated cities. Silicon has been used to increase the anodes’ capacity, to readily react with the Li, forming a Li-rich alloy and decreasing the Li excess at the graphite’s surface.

Lithium sulphur batteries (LSBs) are one of the most promising future replacements to layered oxide LIBs. Even though the technology for producing this battery type has yet to mature, these batteries allow for theoretical specific capacity values far superior to the layered oxide LIBs here reviewed – elemental sulphur (S${}_{8}$) cathodes have a theoretical capacity of 1675 mAh·g${}^{-1}$[121] and may be combined with lithium metal as the anode, which possesses a theoretical specific capacity of 3830 mAh·g${}^{-1}$[121]. Specific energy density as high as 2600 Wh·kg${}^{-1}$ has been reported [121,122] (a reference value for LIBs is <600 Wh·kg${}^{-1}$[121]). Other advantages of LSBs are their environmental friendliness, associated with their nontoxicity [123], and low cost [122]. However, like Si anodes, enhanced capacity and cycle life have been difficult to achieve due to large irreversibilities during (de)-/lithiation and other severe damage mechanisms. The insulating nature of sulphur, the solubility of discharge products in liquid electrolytes, and poor sulphur utilization have been identified [123] as key barriers against the development of Li-S cells. He et al. [122] highlight large volume expansion of sulphur, poor electronic and ionic conductivity of lithium sulfide and sulfur, and poor kinetics of lithium polysulfide redox during the conversion reaction as main limitations. Interestingly, LSBs share a significant amount of weaknesses with Si anodes. For instance, the large volume expansions associated with cycling also lead to particle pulverization, and side reactions between the sulphuric cathode and organic liquid electrolytes also occur [123]. Accordingly, they also share some similarities between strategies to mitigate those issues: nanostructured composite cathodes based on carbon and other types of conducting materials have been designed to improve electronic and ionic conductivity, as well as the stability of sulphur cathodes [124]. LSBs still need significant improvements to become serious alternatives to today’s LIBs – the energy density typically achieved, for practical applications, is only 350 Wh·kg${}^{-1}$ [125]-and technological improvements in the design of Si anodes may help their development, inspiring new and more successful strategies.

Presently, all EV batteries use liquid-state electrolytes [126] composed of organic solvents, LiPF${}_{6}$ (or a similar lithium conductive salt) and various additives [127]. The first two components ensure ionic transfer [28]. Additives are required to further enhance the performance of the battery in terms of safety, charge/discharge kinetics, or mechanical integrity. Due to the flammability of these electrolytes, safety is the main motive for developing new additives, an ongoing research topic for liquid electrolyte LIBs, and one of the key ingredients of a (successful) EV battery, which naturally leads to secrecy surrounding the actual chemical composition of electrolytes. The experimental works here reviewed confirm this reality.

To further reduce the dendrite formation and increase the capacity of the cell by safely using Lithium metal as the anode allowing for the use of these sulphuric high-capacity cathodes such as those containing sulphur (S${}_{8}$), all-solid-state batteries are being developed. Many of the leading manufacturers are looking to build the first EV powered by an all-solid-state battery [128,129,130,131,132], despite apparent disbelief on the technology by some stakeholders (see, e.g., [133]). With different release dates announced—ranging from 2021 (Toyota [128]) to 2030 (Volkswagen [130])—they all agree on the advantages brought by the technology: less cost, longer range, higher charge rates, and low safety concerns, especially compared to liquid electrolyte solutions. However, the solid-state battery effort to replace the Li-ion battery, overcoming safety issues, is plagued by four main bottlenecks: (1) slow kinetics of ion diffusion in solid-state electrolytes, and the transport of ions across the solid-solid interfaces; (2) chemical instabilities at Li metal-solid electrolyte and high voltage cathode-solid electrolyte interfaces; (3) local mechanical and structural instabilities in solid-state electrolytes that fail to resist lithium dendrites and compromise safety; (4) the necessity of renewing the existing Li-ion assembly lines and equipment, which is an additional impediment for fast commercialization of all currently available all-solid-state solutions. This effort requires a new approach eventually involving an amorphous electrolyte that is homogeneous allowing for a 2D lithium plating on a Li-anode [134] and involving a ferroelectric electrolyte that allows for electrostatic charge transfer within the cell as well as an additional energy storage [135]. These latter families of electrolytes allow for novel cell architectures [135,136,137].

## 5. Conclusions

We have reviewed all the major trends in Li-ion batteries technologies used in EVs. We concluded that only five types of cathodes are used by most EV companies, Lithium-Cobalt Oxide (LCO), Lithium-Manganese Oxide (LMO), Lithium Nickel-Manganese-Cobalt oxide (NMC), Lithium Nickel-Cobalt-Aluminum oxide (NCA), and Lithium-Iron Phosphate (LFP). From the latter, most of the companies use NMC as the Nickel allows the energy density to increase, the Manganese reduces the cost, and the Cobalt makes the structure stable inhibiting the release of oxygen. We have detailed the important features of each type of cathode, such as their crystal structure, Lithium hoping mechanisms, and strategies as for example doping and composite synthesis. Most of the Li-ion batteries anodes are graphite-based. Anodes were reviewed in detail, especially the recent efforts to add silicon to graphite in order to avoid dendrite growth and to increase the energy density while avoiding failure due the silicon swell/shrink upon lithiation/delithiation. The electrolyte used is a liquid/gel flammable solvent usually containing LiFeP${}_{6}$ salt. The latter makes the battery and battery pack unsafe and drives the research and development to replace it by a solid-state electrolyte.

## Figures and Tables

**Figure 1 molecules-26-03188-f001:**
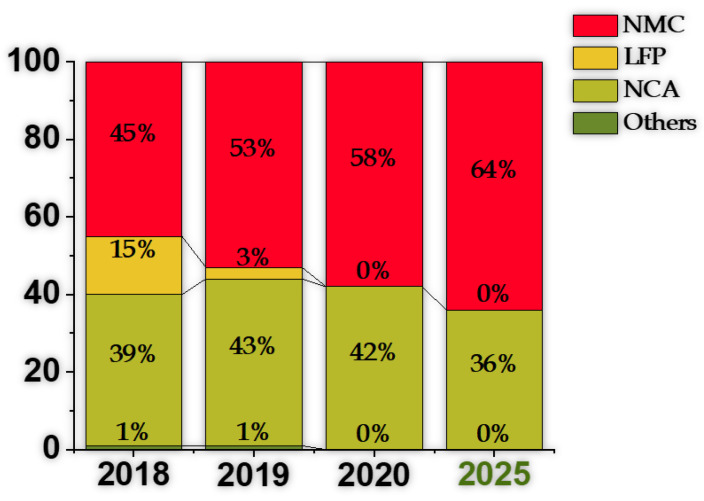
Adoption rate per chemistry in electrical vehicle (EV) battery Market. Adapted from the source [9,10].

**Figure 2 molecules-26-03188-f002:**
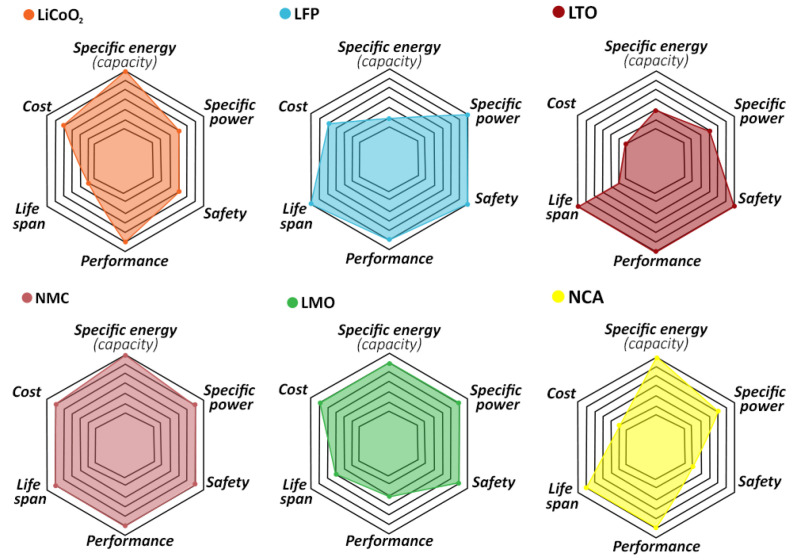
Comparison between different types of cathodes. Adapted from [30].

**Figure 3 molecules-26-03188-f003:**
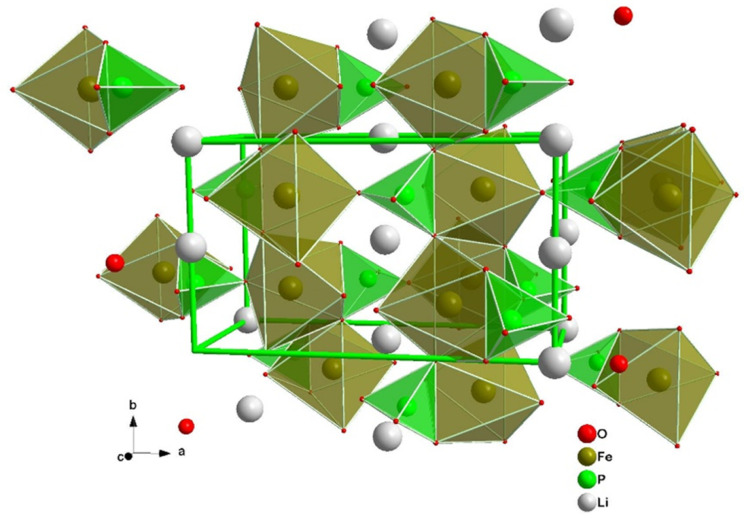
Orthorhombic crystal structure of LiFePO${}_{4}$ (Olivine), space group Pnma.

**Figure 4 molecules-26-03188-f004:**
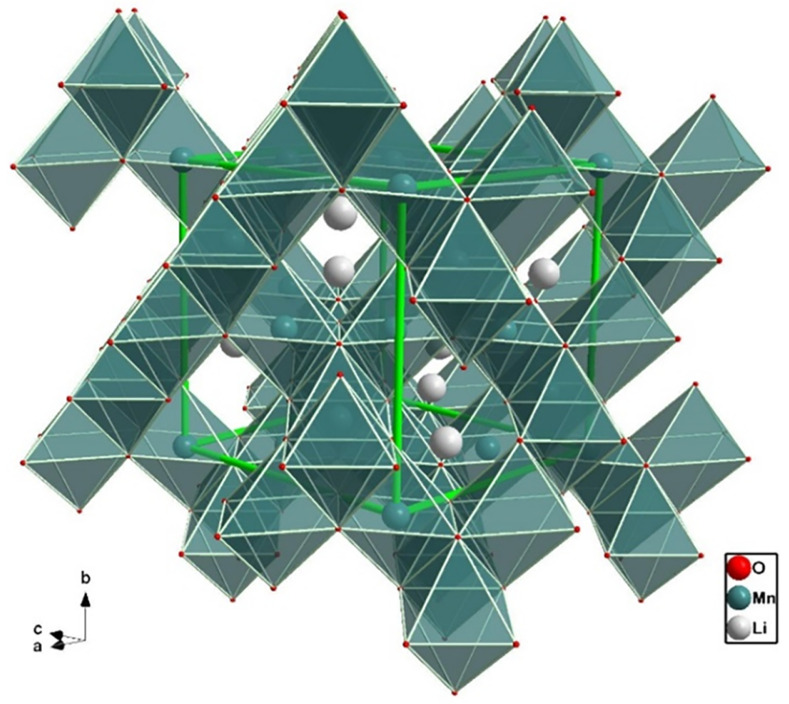
Cubic (spinel) structure of LiMn${}_{2}$O${}_{4}$. Space group: Fd-3m.

**Figure 5 molecules-26-03188-f005:**
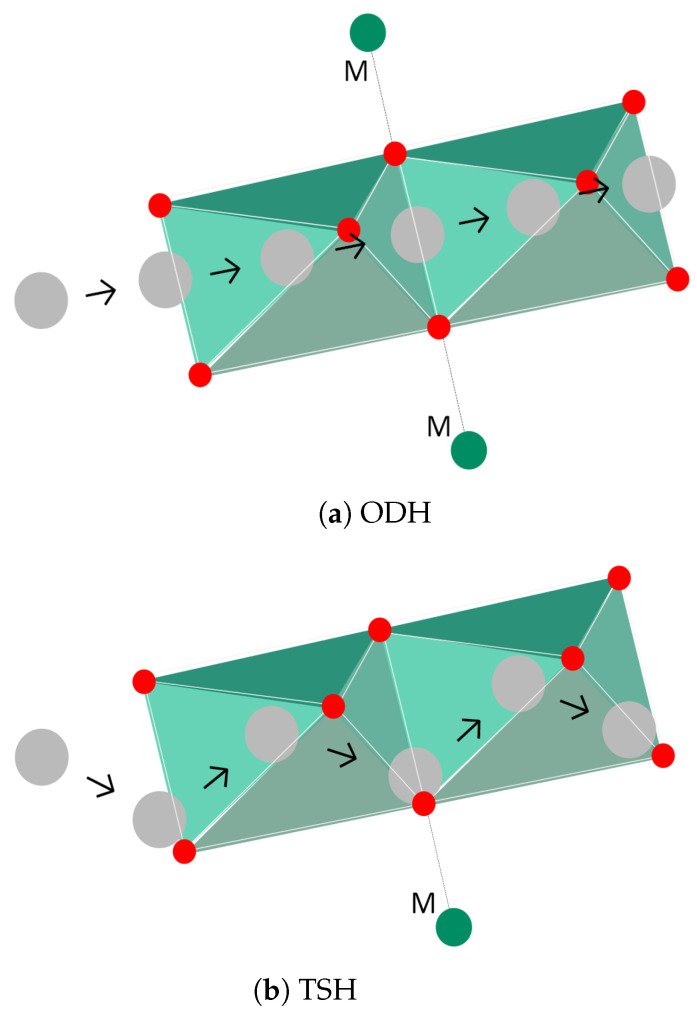
Li-ion diffusion mechanisms in layered oxides: oxygen dumbbell hopping (ODH) and tetrahedral site hopping (TSH).

**Figure 6 molecules-26-03188-f006:**
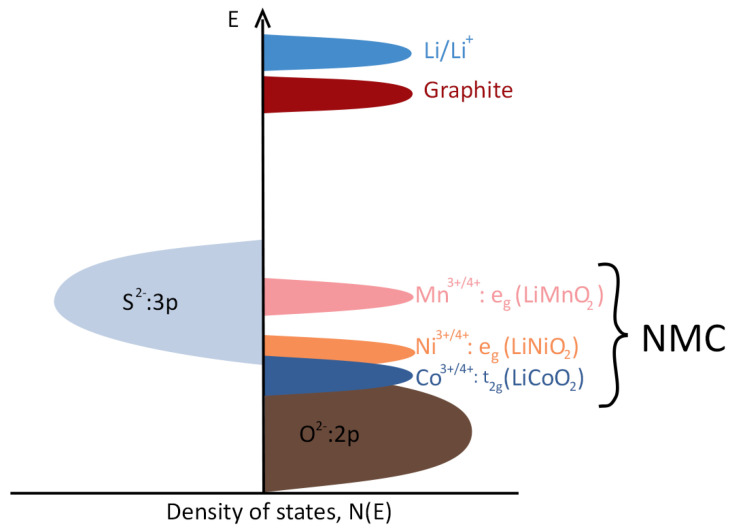
Schematic representation of the density of states of Li${}_{2}$S, graphite, LiMnO${}_{2}$, LiNiO${}_{2}$, LiCoO${}_{2}$, and NMC. Adapted from [25].

**Figure 7 molecules-26-03188-f007:**
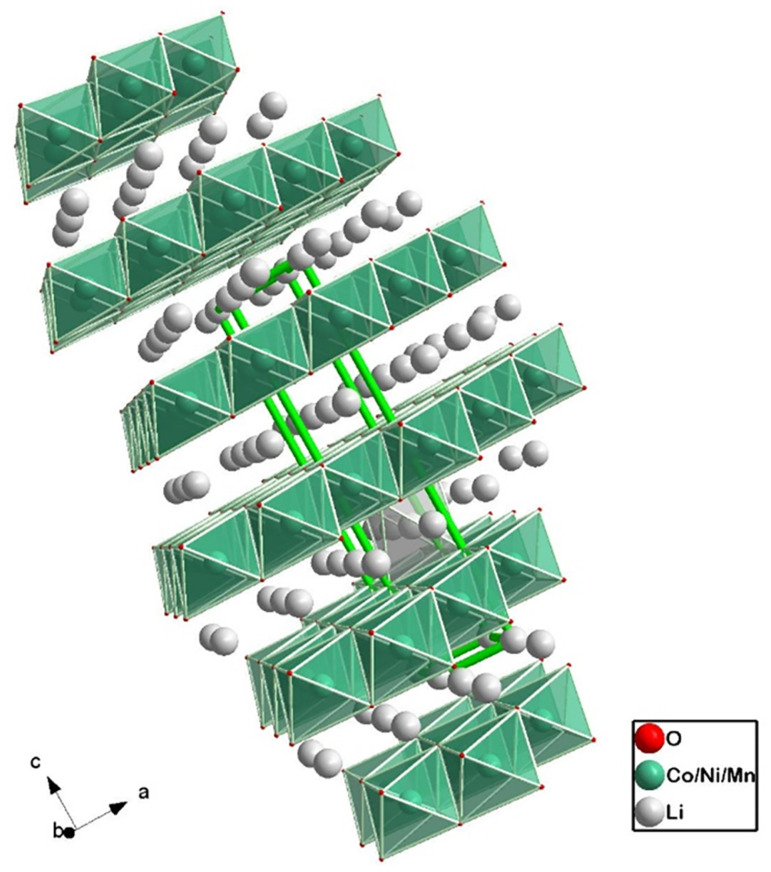
Rhombohedral (layered) structure for NMC622. Space group: R-3m.

**Figure 8 molecules-26-03188-f008:**
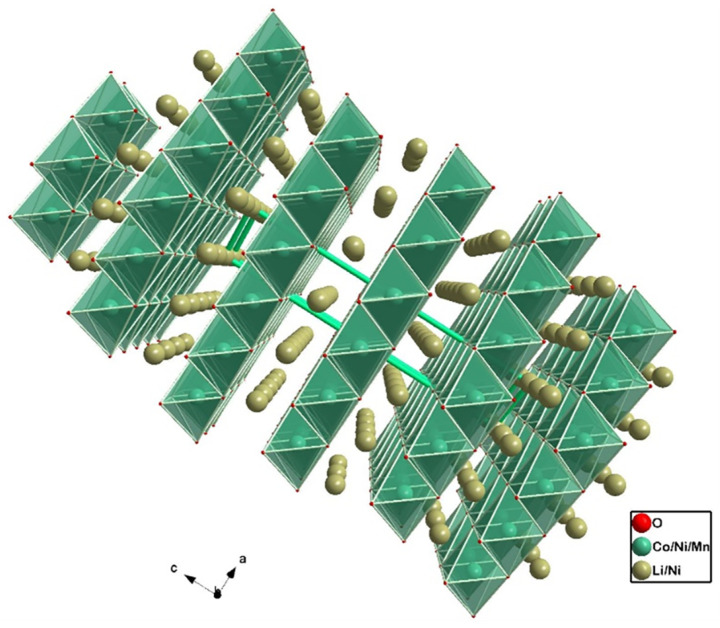
Rhombohedral (layered) structure for NMC811. Space group: R-3m.

**Figure 9 molecules-26-03188-f009:**
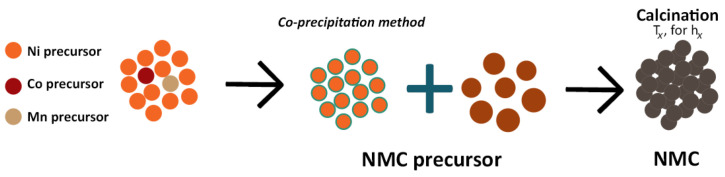
NMC cathodes synthesis: NMC secondary particles (precursor) obtained by co-precipitation of transition metal precursors followed by calcination for h${}_{x}$ hours at fixed temperature T${}_{x}$. Adapted from [59].

**Figure 10 molecules-26-03188-f010:**
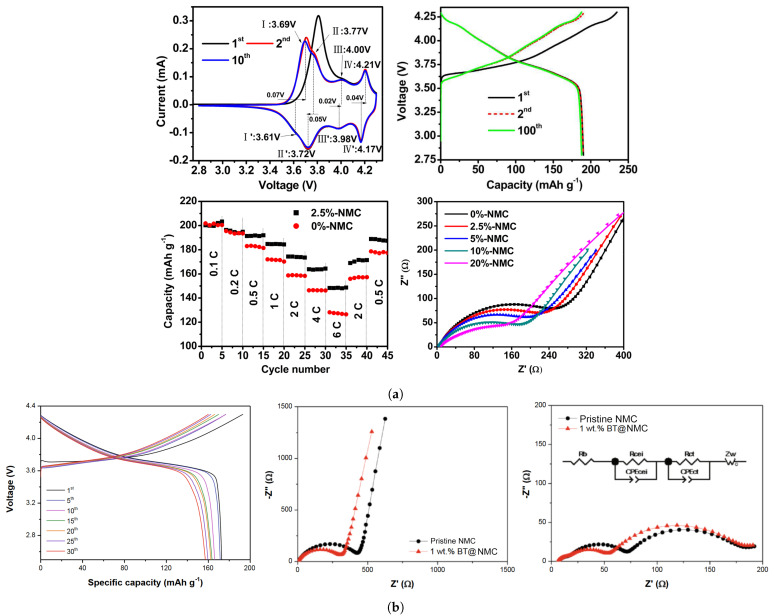
Electrochemical curves for surface coated NMC cathodes. (**a**) Effect of 2.25% carbon coating on NMC: (**Top**) cyclic voltammetry, charge/discharge curves, (**Bottom left**) rate performance. (**Bottom right**) Impedance of NMC cathodes coated with different percentages of carbon. Reproduced with permission [54]. (**b**) Charge/discharge curves for 1 wt% BT@NMC electrodes at 0.1C, Nyquist plots for pristine and 1 wt% BT@NMC after (**center**) first cycle and (**right**) after 100 cycles. Reproduced with permission [72].

**Figure 11 molecules-26-03188-f011:**
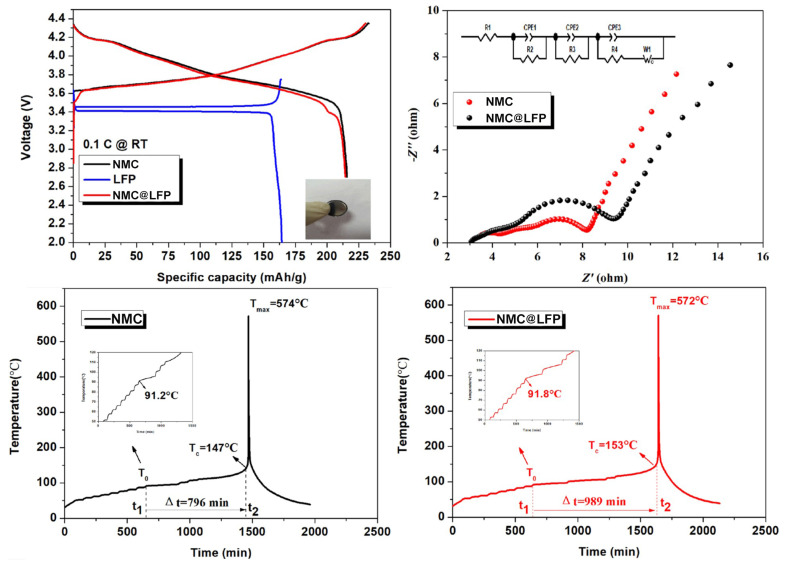
(**Top**) Charge/discharge curves for NMC, LFP, and NMC@LFP composite cathodes; Nyquist plots for NMC and NMC@LFP cathodes; (**Bottom**) Thermal runaway curves for full cells (**left**) NMC || Graphite and (**right**) NMC@LFP || Graphite. Reproduced with permission [53].

**Figure 12 molecules-26-03188-f012:**
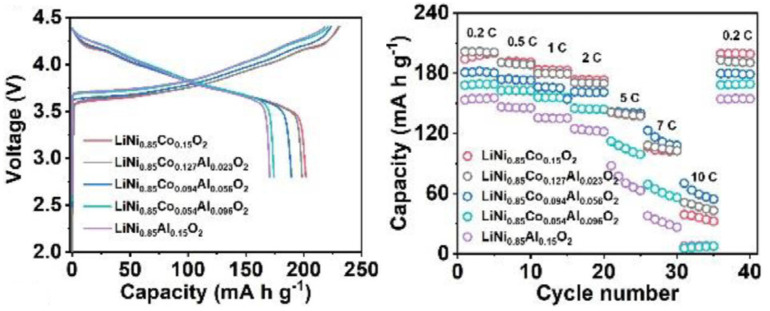
Charge/discharge and rate capacity curves for Al-doped LiNi${}_{0.85}$Co${}_{1.5-x}$Al${}_{x}$O${}_{2}$ cathodes. Reproduced with permission [58].

**Figure 13 molecules-26-03188-f013:**
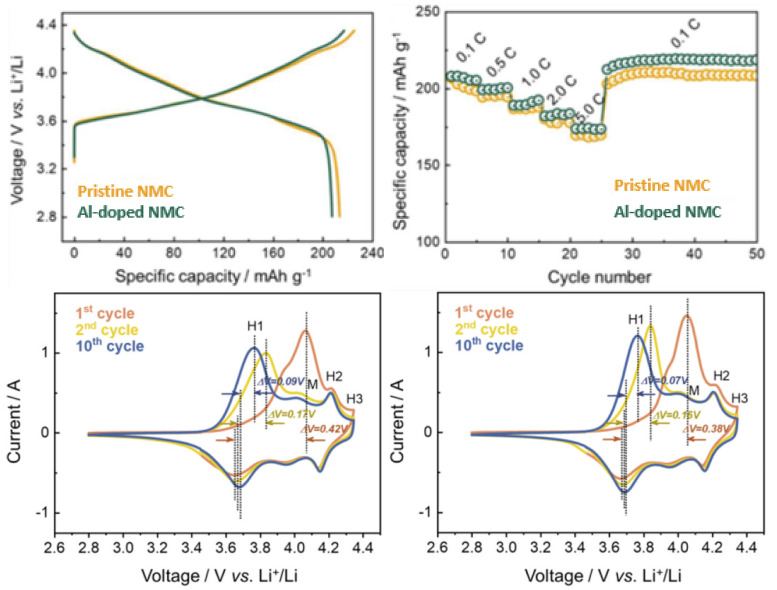
(**Top**) Charge/discharge and specific capacity for different rates for pristine and Al-doped LiNi${}_{0.8}$Mn${}_{0.15}$Co${}_{0.05}$O${}_{2}$, (**Bottom**) cyclic voltammetry curves for (left) pristine and (right) Al-doped LiNi${}_{0.8}$Mn${}_{0.15}$Co${}_{0.05}$O${}_{2}$. Reproduced with permission [57].

**Figure 14 molecules-26-03188-f014:**
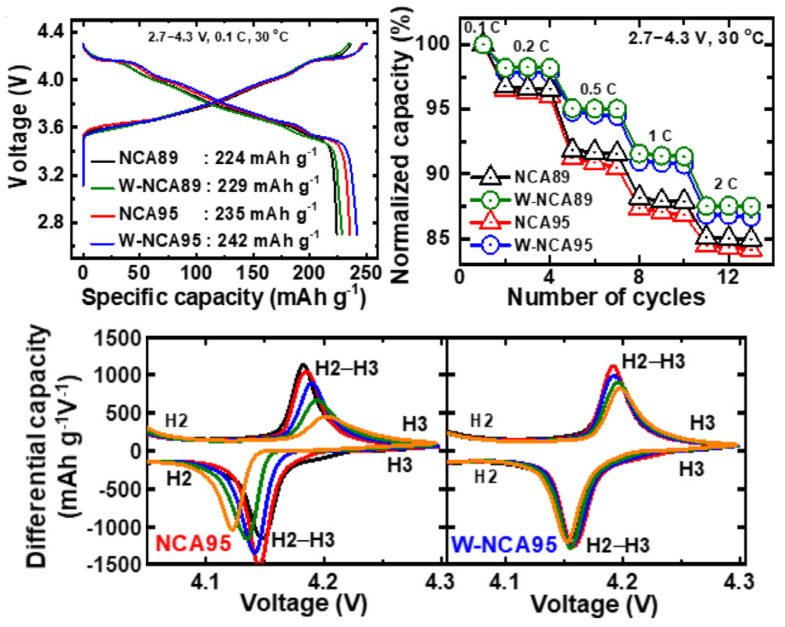
Electrochemical curves for pristine and W doped cathodes: (**Top**) Charge/discharge curves and rate capacity for Ni${}_{0.95}$Co${}_{0.04}$Al${}_{0.01}$ and Ni${}_{0.885}$Co${}_{0.1}$Al${}_{0.015}$, (**Bottom**) cyclic voltammetry for Ni${}_{0.95}$Co${}_{0.04}$Al${}_{0.01}$. Reproduced with permission [55].

**Figure 15 molecules-26-03188-f015:**
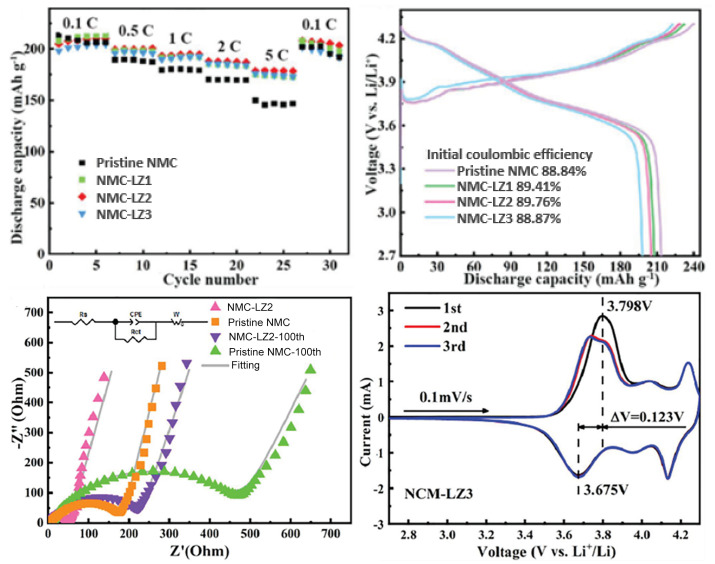
La${}_{2}$Zr${}_{2}$O${}_{7}$ coated and Zr doped NMC cathode: (**Top left**) rate capacity and (**Top right**) charge/discharge curves, (**Bottom left**) Nyquist plots and (**Bottom right**) cyclic voltammetry curves. Graphs for pristine and different coating/doping wt% samples (LZ${}_{x}$, x in wt%). Reproduced with permission [56].

**Figure 16 molecules-26-03188-f016:**
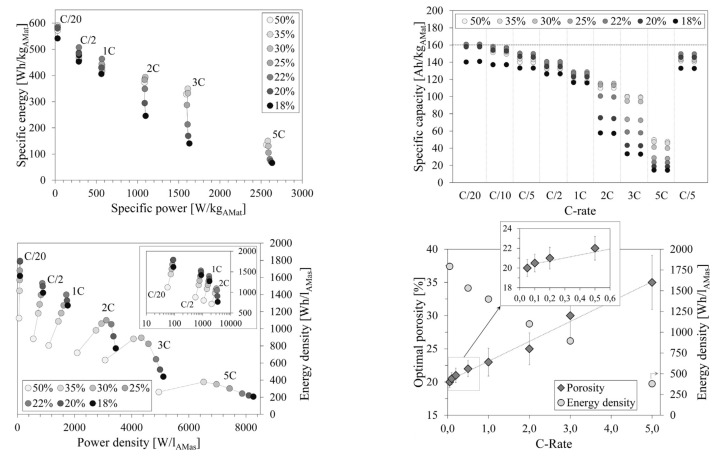
(**Top**) Porosity effect on NMC (LiNi${}_{0.33}$Co${}_{0.33}$Mn${}_{0.33}$O${}_{2}$) cathodes: (left) specific energy vs. specific power and (right) specific capacity vs. C-rate; (**Bottom**) Ragone plots between C/20 and 5C: (left) energy density vs. power density (volumetric evaluation) and (right) optimal porosity to achieve the maximum energy density. Reproduced with permission [70].

**Figure 17 molecules-26-03188-f017:**
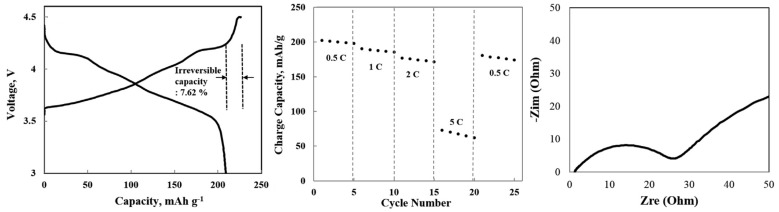
Charge/discharge curves, rate capacity, and initial Nyquist plot for porosity controlled LiNi${}_{0.91}$Co${}_{0.06}$Mn${}_{0.03}$O${}_{2}$ NMC. Reproduced with permission [59].

**Figure 18 molecules-26-03188-f018:**
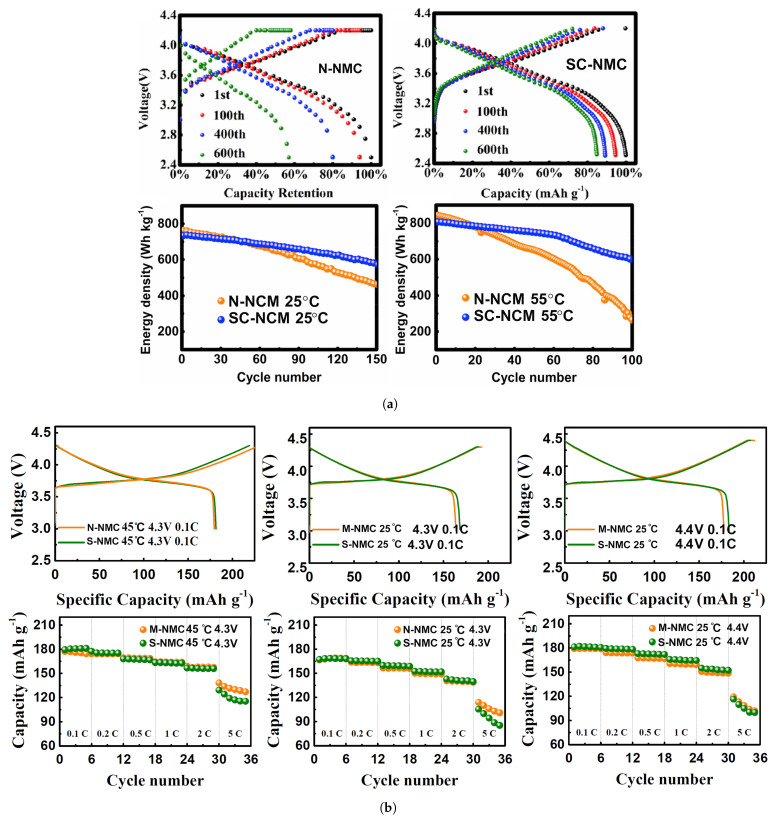
Electrochemical capability comparison between Single-Crystalline (SC-) and Normal (N-)/ Secondary (S-) and
Monocrystalline (M-) NMC cathodes. (**a**) Charge/discharge curves and energy density at 0–100 cycle at 25 °C and 55 °C for
a half-cell. Reproduced with permission [60]. (**b**) Charge/discharge curves and rate capacities for different temperatures
and charging cut-off voltage for both S- and M- samples. Reproduced with permission [71].

**Figure 19 molecules-26-03188-f019:**
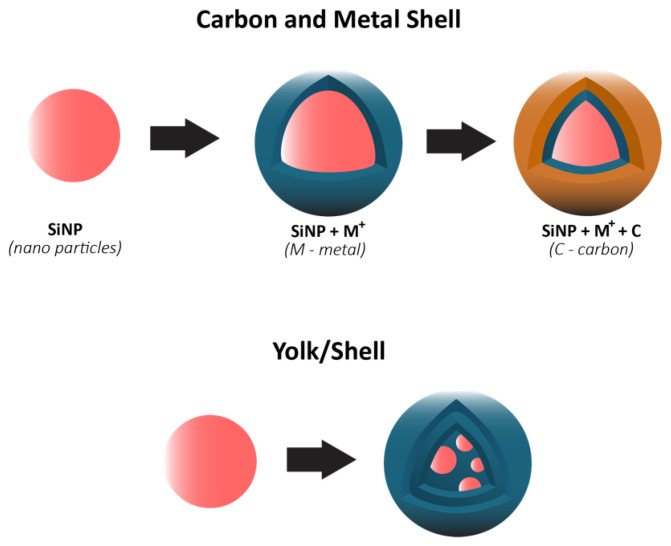
(**Top**) Carbon shell design, and (**Bottom**) Yolk/Shell design strategies for composite Si anodes. Adapted from [109].

**Figure 20 molecules-26-03188-f020:**
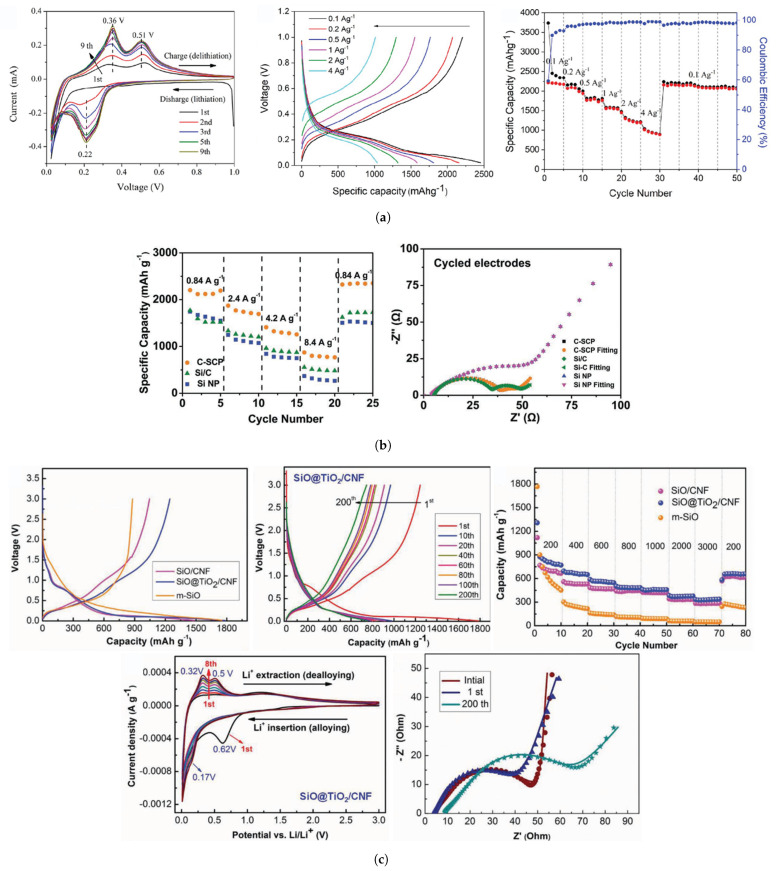
Composite Si/M anodes (M - metal) obtained with a M protective shell and a Si core. All figures reused with
permission. (**a**) Charge/discharge curves, rate capability, and cyclic voltammetry for a Si@C-Cu anode. Reproduced
with permission [111]. (**b**) Rate capability and Nyquist plots comparison between Si NP, Si@Cu (Si/C), and Si@C-Cu
(C-SCP) anodes. Reproduced with permission [108]. (**c**) Charge/discharge curves comparison between SiO@TiO_2_/carbon
nanofiber (SiO@TiO_2_/CNF), SiO/CNF, and ball milled SiO (m-SiO), and cyclic voltammetry curves and Nyquist plots for
SiO@TiO_2_/CNF. Reproduced with permission [110].

**Figure 21 molecules-26-03188-f021:**
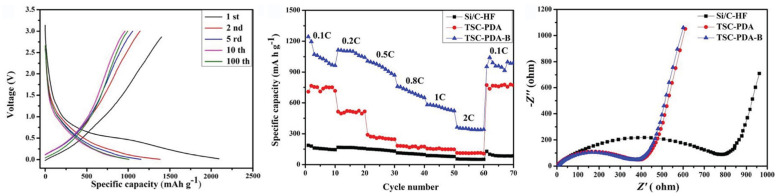
Yolk-Shell SiC composite anodes-single carbon layer (Si/C-HF), double carbon layer with single Si core (TSC-PDA), and double carbon layer with broken Si cores (TSC-PDA-B): Charge/discharge curves for TSC-PDA-B, rate capability, and Nyquist plots, for all anode types. Reproduced with permission [109].

**Figure 22 molecules-26-03188-f022:**
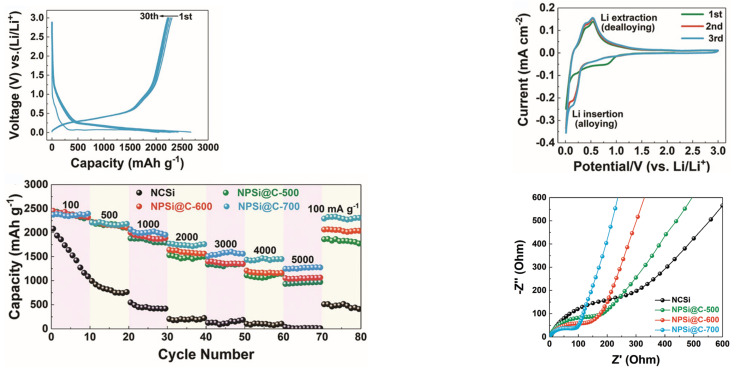
(**Top left**) Charge/discharge curves and (**Top right**) cyclic voltammetry curves, (**Bottom left**) rate capability, and (**Bottom right**) Nyquist plots for distinct NP-Si@C anodes with different etching temperatures. Reproduced with permission [107].

**Figure 23 molecules-26-03188-f023:**
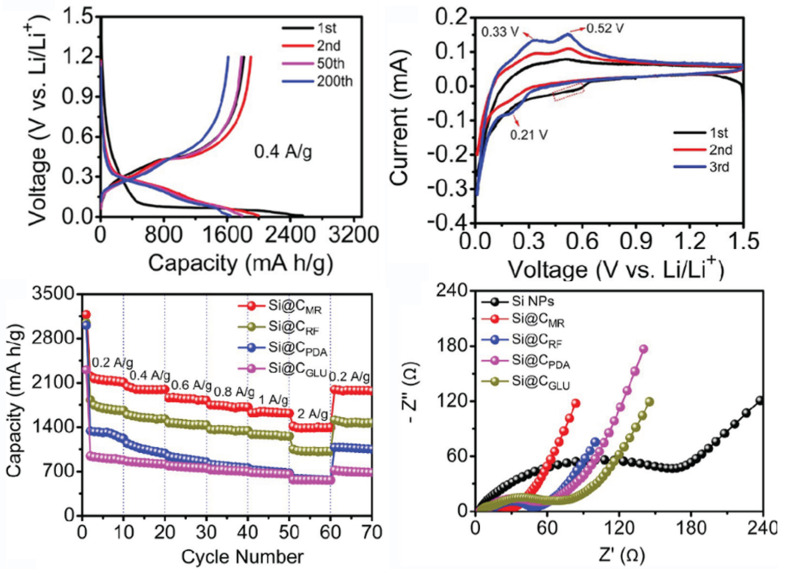
(**Top**) Electrochemical curves for Si/C anode obtained using melamine resin as carbon precursor (Si@C${}_{MR}$): (**left**) Charge/discharge curves and (**right**) cyclic voltammetry curves. (**Bottom**) Electrochemical curves for Si/C anodes obtained via distinct carbon precursors: (**left**) rate capability and (**right**) Nyquist plots. Reproduced with permission [106].

**Figure 24 molecules-26-03188-f024:**
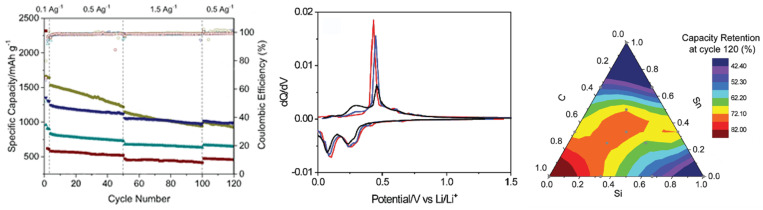
Specific capacity, cyclic voltammetry, and capacity retention for Si/C/Sn composite anode [116]. Copyright Wiley-VCH GmbH. Reproduced with permission.

**Figure 25 molecules-26-03188-f025:**
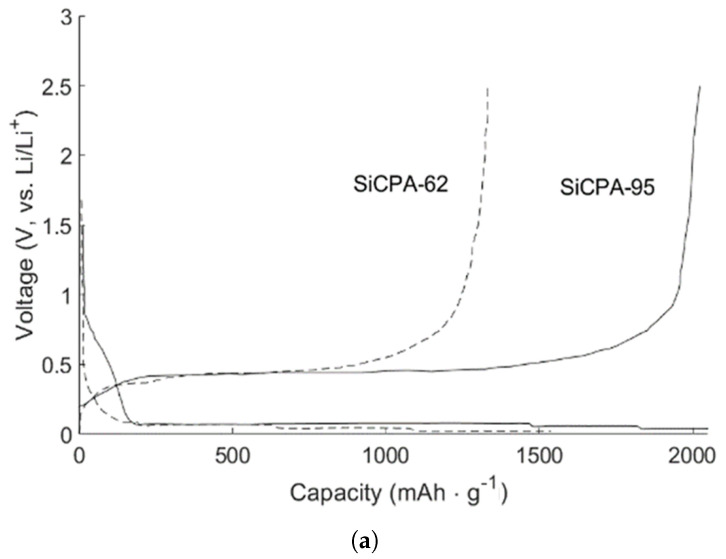
Si anodes enhanced by alternative techniques. (**a**) Charge/discharge curves for Si anodes binded by Polyamid films. Reproduced with permission [117]. (**b**) Charge/discharge curves, rate capability for Si anodes binded by a networked structure incorporating hard and soft LiPAA/LiPN phases. Reproduced with permission [118]. (**c**) Charge/discharge curves, and rate capability of half-cells with Si anode and high salt concentration liquid electrolyte. Reproduced with permission [120].

**Table 1 molecules-26-03188-t001:** Experimental capacity, plateau voltage, thermal runaway temperature, and cycle life for different cathodes (reference values [30]).

Cathode Type	Formula (General)	Experimental Capacity (mAh·g${}^{-1}$) ${}^{1}$	Plateau Voltage (V vs. Li${}^{0}$/Li${}^{+}$)	Thermal Runaway (${}^{\circ }$C)	Cycle Life (No. of Cycles) ${}^{2}$
Li Cobalt Oxide (LCO)	LiCoO${}_{2}$	150	4.3–3.8	150	500–1000
Li Manganese Oxide (LMO)	LiMn${}_{2}$O${}_{4}$	120–130	4.3–3.8	250	300–700
Li Nickel-Manganese-Cobalt oxide (NMC)	LiNi${}_{x}$Mn${}_{y}$Co${}_{z}$O${}_{2}$ (*x* + *y* + *z* = 1)	150	4.3–3.7	210	1000–2000
Li Nickel-Cobalt-Aluminum oxide (NCA)	LiNi${}_{x}$Co${}_{y}$Al${}_{z}$O${}_{2}$ (*x* + *y*+*z* = 1)	175	4.3–3.5	150	500
Li-Iron Phosphate (LFP)	LiFePO${}_{4}$	160–170	3.3	270	>2000

${}^{1}$ Cutoff voltage at 2 V; ${}^{2}$ Significantly dependent on specific application and environment. Some cathodes reach cycle lives far greater than the displayed values, for example, Yuasa’s LEV50 battery’s LMO cathode retains 80% capacity after 5500 charge/discharge cycles.

**Table 2 molecules-26-03188-t002:** Main cathodes, excluding Lithium Nickel-Manganese-Cobalt (NMC), used in EVs. and their main characteristics [28,32,33,34,35,36,37,38,39]. Also available on the ALBATTS project website [40].

Cathode Type	Ratios (R) or Cell Designation (S) ${}^{3}$	Manufacturer	No. of Cells (Series, Parallel)	EV Model	Specific Energy (Wh/kg)	Energy (Usable) (kWh)	Range ${}^{4}$ (km)
Li-Nickel- Cobalt- Aluminum oxide (NCA)	18650 (S) 2170 (S)	Panasonic	8256 (s96p86) 4416 (s96p46)	Tesla Model S, Tesla Model X, Tesla Model 3	162 168	102.4 (98.4) 80.5 (76)	593, 487, 530
Li-Manganese Oxide (LMO)	-	Yuasa	80	Citroen Zero (LEV50 battery)	107	14.5	150
Li-Cobalt Oxide (LCO)	-	LG Chem	96	Smart Fortwo e	150–200	17.6 (17.2)	127
Lithium-Iron Phosphate (LFP)	-	Elektrofahrzeu- ge Stuttgart CATL BYD Blade	- - - 102	Iridium E_Mo- bil Tesla Model 3 ${}^{5}$ BYD Han EV	90–120 125 -	106 106 65	400 400 506

${}^{3}$ Ratios–Metal proportions used for NMC, as explained above; Cell designation–refers to a cell’s dimension. ${}^{4}$ Combined. WLPT values. ${}^{5}$ Chinese market.

**Table 3 molecules-26-03188-t003:** NMC cathodes used in EVs. and their main characteristics [28,32,33,34,35,36,37,38,39]. Also available on the ALBATTS project website [40].

Cathode Type	Ratios (R) or Cell Designation (S)	Manufacturer	No. of Cells (Series, Parallel)	EV Model	Specific Energy (Wh/kg)	Energy (Usable) (kWh)	Range ${}^{6}$ (km)
	532 (R)	Nissan CATL Envision AESC	288 216 (s108p2) 192 (s96p2)	Nissan Leaf e+ Peugeot e-208, Opel Corsa-e Nissan Leaf	- 140 130	62 50 (46) 39.5 (36)	385 349, 336 270
	333 (R)	Samsung SDI	264 (s88p3)	Volkswagen e-Golf	103	35.8 (32)	232
	721 (R)	LG Chem	192 (s96p2)	Renault ZOE	168	54.7 (52)	232
Li-Nickel Manganese Cobalt oxide (NMC)	622 (R)	Samsung SDI SK Innovation LG Chem	96 (s96p1) 294 (s98p3) 168 (s84p2) 176 (s88p2) 294 (s98p3) 384 (s96p4) 396 (s198p2) 432 (s108p4) 288 (s96p3)	BMW i3 Kia e-Soul, Kia e-Niro Volkswagen e-UP, Seat Mii Electric, Skoda CITIGo-e Hyundai Ioniq-e Hyundai Kona-e Mercedes-Benz EQC Porsche Taycan Jaguar I-Pace Audi e-tron 55 Quattro Chevrolet bolt	152 148 148 112.4 149 130 148 149 136 143	42.2 (37.9) 67.5 (64) 36.8 (32.3) 40.4 (38.3) 67.5 (64) 85 (80) 93.4 (83.7) 90 (84.7) 95 (86.5) 68	293 451, 454 260, 256, 265 310 447 417 333 470 402 417

${}^{6}$ Combined. WLPT values.

**Table 4 molecules-26-03188-t004:** Commercially available anodes and their main features.

Anode Type	Application	Voltage	Capacity	Specific Energy	Cycle life
		(V vs. Li${}^{0}$/Li${}^{+}$)	(mAh·g${}^{-1}$)	(Wh·kg${}^{-1}$)	
Graphite (C)	Most commercially available batteries	0.15–0.25	372	100–156	2000
Lithium-Titanium Oxide Li${}_{4}$Ti${}_{5}$O${}_{12}$ (LTO)	LFP batteries	1.5	175	50–80	3000–7000
Silicon	Nanowire (SiNW) Amprius Technologies: Airbus Zephyr S pseudo satellite HAPS Military vehicles	0.4	4200 (Silicon) 3579 (SiNW)	435 (Amprius)	>2000 (SiNW)

**Table 5 molecules-26-03188-t005:** Composite Si anodes proposed in recent years: cycle voltage, initial discharge capacity, and ICE.

Anode Type	Cycle Voltage (V)	Initial Discharge Capacity (mAh· g${}^{-1}$)	ICE (%), No of Cycles for >99%
SiNP [105]	0.01–1.5	2914.3	∼80
SiONP [110]	-	1755	50.4
Si@C-CNT-Cu [111]	-	∼2341	∼88, 10
Si@C/CNTs@GS [105]	0.01–1.5	2533.3	87.6
SiO@TiO${}_{2}$/CNF [110]	-	1782	69.8, 11
Si@SiO${}_{2}$@C [109]	0.05–3	2108	71
NP-Si@C [107]	-	2305.9	86.56
C-SCP [108]	-	3346	81
Si@CMR [106]	0.01–1.2	1834.2	71, 5
Si@CRF [106]	0.01–1.2	∼3100	-
Si@CPDA [106]	0.01–1.2	∼1834	-
Si@CGLU [106]	0.01–1.2	∼2800	-
Sn${}_{33}$Si${}_{33}$C${}_{33}$ [116]	-	1499.5	86.7
SiCPA-62 [117]	0.005–2.5	1247	86
SiCPA-95 [117]	0.005–2.5	2350	87
Si@N-P-LiPN [120]	0.005–1.2	3614	93.18
Si@Fe${}^{3+}$-PDA/PAA [119]	0.01–2	4000	-

**Table 6 molecules-26-03188-t006:** Composite Si anodes proposed in recent years: discharge capacity after *x* cycles and rate performance.

Anode Type	Discharge Capacity (mAh· g${}^{-1}$), after *x* Cycles	Rate Performance
SiNP [105]	<500, 130	-
SiONP [110]	<158, 200	-
Si@C-CNT-Cu [111]	1500, 900	2168, 1837, 1577, 1236, 942 (0.2, 0.5, 1, 2, 4 /A·g${}^{-1}$)
Si@C/CNTs@GS [105]	1524.3, 130	1910, 1630, 1430, 1000, 1530 (0.2, 0.4, 0.8, 1.6, 0.1 /A·g${}^{-1}$)
SiO@TiO${}_{2}$/CNF [110]	∼760, 200	875, 696, 588, 502, 460, 384, 338, 713 (0.2, 0.4, 0.6, 0.8, 1, 2, 3, 0.2 /A·g${}^{-1}$)
Si@SiO${}_{2}$@C [109]	113, 200	1243, 1050, 870, 650, 520, 340, 960 (0.1, 0.2, 0.5, 0.8, 1, 2, 0.1 /A·g${}^{-1}$)
NP-Si@C [107]	2126.2, 120	∼2180, 1990, 1750, 1530, 1490, 1271.3, 2287.3 (0.5, 1, 2, 3, 4, 5, 0.1 /A·g${}^{-1}$)
C-SCP [108]	1050, 1000	2202.6, 1870.4, 1408.3, 873.7, 2323.3 (0.84, 2.4, 4.2, 8.4, 0.84 /A·g${}^{1}$)
Si@CMR [106]	∼1614.6, 200	2126.7, 1993.6, 1851.2, 1741.8, 1628.7, 1994.4 (0.2, 0.4, 0.6, 0.8, 1, 2, 0.2 /A·g${}^{-1}$)
Si@CRF [106]	1064.5, 200	∼1700, 1570, 1450, 1380, 1320, 1050, 1500 (0.2, 0.4, 0.6, 0.8, 1, 2, 0.2 /A·g${}^{-1}$)
Si@CPDA [106]	880.1, 200	∼1200, 1000, 875, 780, 730, 630, 1050 (0.2, 0.4, 0.6, 0.8, 1, 2, 0.2 /A·g${}^{-1}$)
Si@CGLU [106]	700.1, 200	∼850, 817, 770, 740, 710, 620, 700 (0.2, 0.4, 0.6, 0.8, 1, 2, 0.2 /A·g${}^{-1}$)
Sn${}_{33}$Si${}_{33}$C${}_{33}$ [116]	-	1165, 965, 809, 822 (0.1, 0.5, 1.5, 0.5 /A·g${}^{-1}$)
SiCPA-62 [117]	∼700, 250	1279.8, 1094.3, 922.1, 729.4 (0.2, 0.5, 1, 2 /A·g${}^{-1}$)
SiCPA-95 [117]	∼1287, 50	1812.4, 1290.6, 920.7, 524.6 (0.2, 0.5, 1, 2 /A·g${}^{-1}$)
Si@N-P-LiPN [120]	∼2159, 100	3859.7, 3533.1, 3199.0, 2620.9, 1598.9, 3417.3 (0.1C, 0.2C, 0.5C, 1C, 2C, 0.2C; 5 cycles step)
Si@Fe${}^{3+}$-PDA/PAA [119]	∼2000, 200	2800, 2100, 1400, 350 (0.2C, 0.5C, 1C, 5C)

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
