# Peer review of "The Latest Trends in Electric Vehicles Batteries"

_molecules, 2021, doi:10.3390/molecules26113188_

Round 1

Reviewer 1 Report

The paper “The latest trends in Electric Vehicles batteries” by R.M. Salgado at al. reviews the major trends in the Li-ion batteries, particularly focusing on the electrodes materials.

The manuscript is interesting and well written and deserves publication. However in my opinion while the cathodes materials are discussed in detail, less is said for the anode materials and not even a paragraph is devoted to the electrolytes. So I suggest to add at least a short paragraph about current used electrolytes and possible solutions to improve their performance and safety.

Author Response

  1. Role of cathodes (vs anodes, electrolyte) - The EV industry has put significant focus on the development of new, "Cobalt-free" cathodes over recent years. This lead to the choice of discussing them in more detail, even though we are aware that it is necessarily the best approach from a more technical point of view.
  2. A paragraph regarding the electrolyte was added to the section Future Prospects due to its connection with the paragraph about "all-solid-state batteries". We aimed to highlight the similarities between liquid electrolytes currently used. The secrecy surrounding additives to improve safety is mentioned, given the significant difficulty in obtaining information about their use in commercial EVs.
  3. Additives and transition to solid-state electrolytes have been identified as solutions to improve the safety feature of secondary batteries. 

Reviewer 2 Report

The article presented is a comprehensive review about the materials (i.e. anode and cathode mainly) used for lithium ion batteries for EV application. The review is well organized and well documented, showing tables comparing all materials as well as evaluation of the market for EV. I would like to include more information about the Lithium sulfur batteries since this technology presents more advantages than lithium ion batteries in the future scenario. Additionally, in the future prospect point, the authors should include the trends in the state of the art for the future design of the new materials for cobalt-free cathode.

Author Response

A new paragraph concerning lithium sulphur batteries was added to Future prospects. Information regarding its advantages over lithium ion batteries in futures scenarios was added, while issues and potential solutions for its development as a commercial alternative were briefly addressed.